# Episodic back-arc spreading centre jumps controlled by transform fault to overriding plate strength ratio

Nicholas Schliffke [1✉], Jeroen van Hunen [1], Mark B. Allen [1], Valentina Magni [2] & Frédéric Gueydan[3]

Spreading centre jumps are a common feature of oceanic back-arc basins. Jumps are conventionally suggested to be triggered by plate velocity changes, pre-existing weaknesses, or punctuated events such as the opening of slab windows. Here, we present 3D numerical models of back-arc spreading centre jumps evolving naturally in a homogeneous subduction system surrounded by continents without a trigger event. Spreading centres jump towards their subduction zone if the distance from trench to spreading centre becomes too long. In particular, jumps to a new spreading centre occur when the resistance on the boundary transform faults enabling relative motion of back-arc and neighbouring plates is larger than the resistance to break the overriding plate closer to trench. Time and distance of spreading centres jumps are, thus, controlled by the ratio between the transform fault and overriding plate strengths. Despite being less complex than natural systems, our models explain why narrow subducting plates (e.g. Calabrian slab), have more frequent and closely-spaced spreading jumps than wider subduction zones (e.g. Scotia). It also explains why wide back-arc basins undergo no spreading centre jumps in their life cycle.

[1] Department of Earth Sciences, Durham University, DH1 3LE Durham, UK. [2] The Centre for Earth Evolution and Dynamics (CEED), University of Oslo, SemSaelandsvei 24, PO Box 1048, Blindern, NO-0316 Oslo, Norway. [3] Géosciences Montpellier, Université Montpellier, place E. Bataillon 34095, Montpellier cedex 5, France. ✉email: nico.schliffke@web.de

The cycle of an oceanic plate from the formation at a sea-floor spreading to consumption at subduction zones is fundamental to plate tectonics[1]. Most subduction zone trenches retreat with respect to the lower mantle, i.e. they migrate towards the subducting plate[2,3]. This 'rollback' exerts tensional stresses to the overriding lithosphere and frequently forms back-arc spreading centres[4,5] that may evolve into small oceanic basins. Formation and location of back-arc spreading centres is controlled by heterogeneities in the buoyancy of subducting plate or strength of the overriding plate: rupturing of the overriding plates is often enhanced by either weak arcs with melt- or fluid-weakening[6,7] or by trench rotation due to subduction of buoyant material[8–10].

While spreading in most back-arc basins ceases after 5–15 Myr of activity[8], others experience long-term spreading and episodic back-arc spreading jumps[8,11,12], when old back-arc spreading centres are abandoned and new ones form closer to the trench. Such jumps have been recorded in South East Asia (e.g. Philippine Sea plate[13], Lau Basin[14,15]) and in relatively narrow subduction zones (e.g. Scotia[16–18], Lesser Antilles[19] and Tyrrhenian Sea[20,21]). The former regions experienced large-scale plate-reorganisation[14] and subduction polarity flips[14,22], possibly causing transient stress field changes that facilitate back-arc spreading jumps. The latter regions, however, show continuous trench retreat (with respect to fixed neighbouring plates and the upper mantle) without necessarily requiring major plate reorganisations, plate rotations, or significant variations in buoyancy or rheology. In those regions, the main back-arc basin characteristics are large-scale bounding transform faults that decouple them from their neighbouring plates, and STEP[2,23,24] ('subduction-transform-edge-propagator') faults that are created at slab edges, thereby tearing through lithosphere at the plate boundary (Fig. 1).

Previous explanations for the occurrence of spreading jumps invoked internal or external triggers, such as interaction between the oceanic slab and the 660-km discontinuity[21], opening of lateral slab windows at depth[9], rearranged mantle flow[11], or inherited weaknesses in the overriding plate[12]. The role of prominent STEP and transform faults in potentially controlling the back-arc spreading jumps have never been constrained.

This work thus focuses on subduction systems exhibiting back-arc spreading jumps without significant plate reorganisation, but instead associated with STEP- and transform faults. We hypothesize that internal feedback mechanisms within a homogeneous subduction system can lead to episodic back-arc spreading centre jumps without any triggering: as back-arc basins open, the transform faults at their edge become longer and induce more resistance. Back-arc spreading centres may lock up as a consequence, and jump towards a new location closer to the trench, thereby reducing the transform fault length. To test this, we investigate the occurrence and characteristics of back-arc spreading jumps in three-dimensional numerical models with homogenous plates.

## Results

**Controls of trench retreat and back-arc spreading.** Back-arc spreading and trench rollback are intrinsically linked. In models with subducting slabs only and no overriding plate, resisting forces are limited to deformation of the slab and mantle, and in that case trench retreat is a function of slab strength[25] and width as well as toroidal mantle flow around slab edges[26]. Adding a strong or thick overriding plate reduces retreat velocities[27,28], and high retreat velocities are most likely for thin overriding plates[28,29], or for those ruptured by back-arc spreading centres[12]. In addition, the presence of neighbouring plates further reduces trench retreat velocities[30]. The strength of interfaces to

neighbouring plates, along which the retreating slab edge must tear through lithosphere, controls trench retreat and curvature[31]: high tear resistance causes strong trench curvature and slow rollback[31]. In contrast, low tear resistance facilitates trench retreat.

The transition from a rigid overriding plate to back-arc spreading requires other processes or weaknesses in addition to extensional stresses from trench rollback alone[9,11]. Melt- or fluid percolation[6,32] or a thin volcanic arc[7] can substantially weaken the overriding plate and support plate rupturing. Moreover, trench rotation localising tensional stresses locally after adjacent continental collision, has shown to trigger back-arc basins formation[9,10]. Mechanisms for opening back-arc basins have been explored in some detail in analogue and numerical models, but modelling the mechanisms for subsequent back-arc spreading jumps has received much less attention.

Tracing back-arc spreading centre jumps across geological timescales is difficult because abandoned back-arc basins are often deformed or destroyed by subduction. Examples of recent inactive back-arc basins that are currently subducting are the Sulu Basin (which opened between 19 and 15 Ma during subduction of the Celebes Sea[8]) and the North Loyalty Basin (opened 44–35 Ma in the New Caledonia subduction zone[8]). However, several active and recent examples do provide data for the length scales and repeat rates of ridge jumps. In the Central Mediterranean subduction zone, the southeastward rollback of the Calabrian slab caused overriding plate extension in the Tyrrhenian Sea since about 14 Ma[21,33] (Fig. 1a). Slab rollback is enabled by a STEP fault in the south[34] Episodic pulses of fast trench rollback[21] coincide with jumps from the extensional Cornaglia Basin to spreading in the Vavilov Basin (4.3–2.6 Ma)[35] and most recently the Marsili Basin (~2–1 Ma)[36,37]. These pulses have been suggested to be caused by slab stagnation on top of the lower mantle[21] and/or opening of slab windows in the previously wider slab[9]. The Scotia subduction zone has retreated eastwards between Antarctica and South America since ~40 Ma[16,17] with two major back-arc spreading centres (Fig. 1b). The present-day configuration was established with the opening of the East Scotia ridge at ~6–8 Ma, close to the eastward retreating trench, and the West Scotia Ridge simultaneously becoming inactive[18,38] (see comment in Supplementary Tables). Other than inherited weaknesses or overriding plate heterogeneities[12], no triggers for this ridge jump have been identified. Although tectonically complexity with local plate rotations, in the Caribbean, an Eocene back-arc spreading jump from the Venezuelan Basin to the Grenada/Tobago Basin has been proposed to be essentially caused by continuous rollback between bounding STEP-faults[19] (Fig. 1c).

**Modelling results**. We present spreading centre jumps in 3D-models with a simple visco-plastic rheology without any compositional strength variation (see Methodology). A narrow oceanic basin (along-strike width varied between 400–1500 km) is situated between two continental plates and subducting below an overriding continental plate (Fig. 2a). To establish gravitationally driven subduction, a slab segment is initially placed in the mantle and between short prescribed transform faults.

Initially, subduction and rollback can occur with little resistance between the pre-existing transform faults in all models due to decoupling of subducting and overriding plate from neighbouring plates (see Supplementary Figures for initial evolution). To study natural plate interaction, we limit the extent of the prescribed transform faults and allow for their spontaneous propagation. Once rollback reaches the end of the prescribed transform faults, further rollback of the subducting slab is only feasible by tearing of lithosphere between subducting and

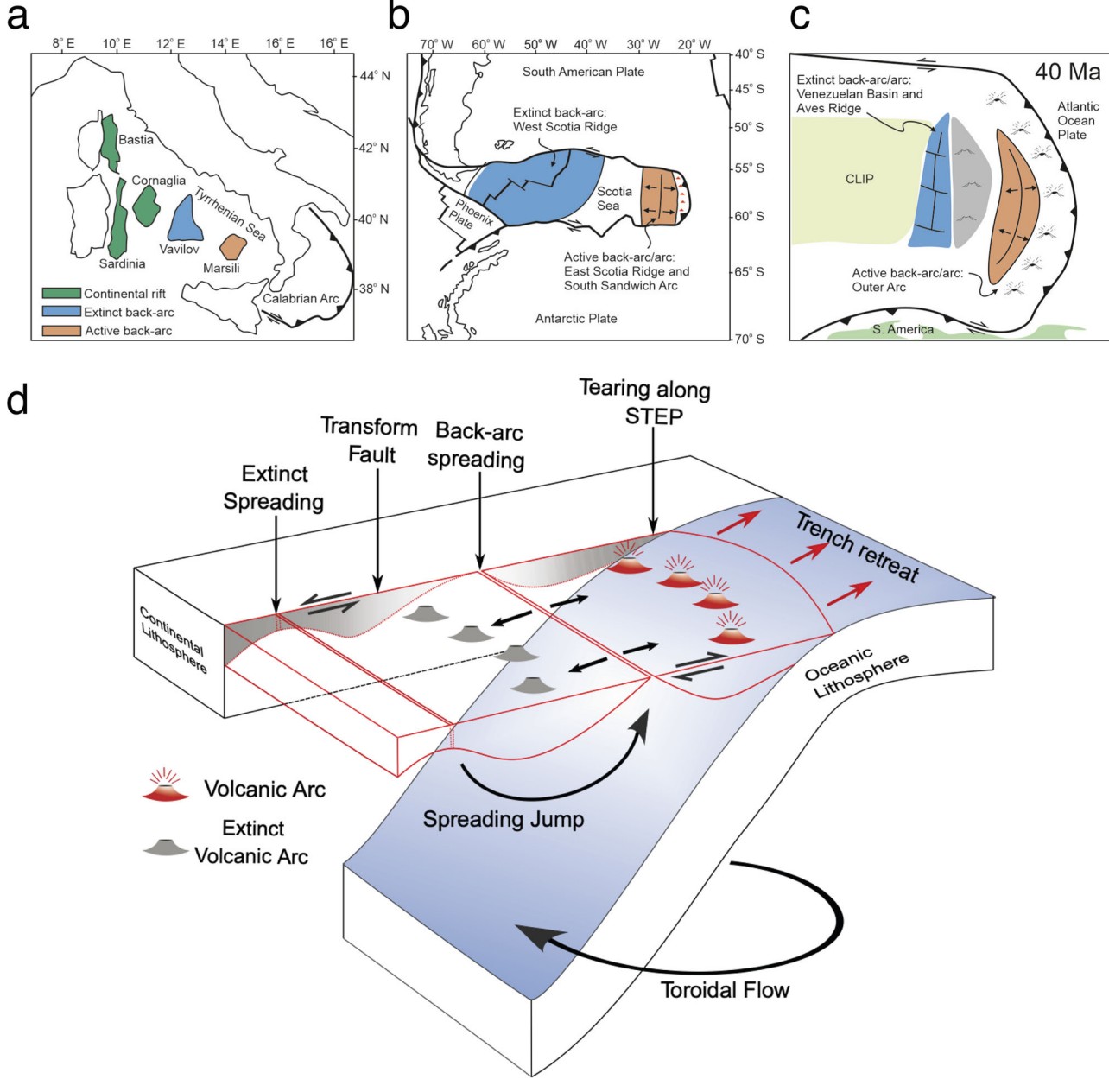

**Fig. 1 Mechanism sketch. a–c** Extinct and active back-arc spreading centre in the narrow Tyrrhenian Sea (adapted from Gueydan et al.[33]), Scotia (adapted from Eagles & Jokat[18], and Lesser Antilles at ~40 Ma (adapted from Allen et al.[19]. All subduction zones retreat within the bounds of STEP- or similar transform faults. **d** Sketch of a spreading centre jump in a retreating subduction zone, bound laterally by transform faults. The subducting slab tears along STEP (subduction-transform-edge-propagator) faults, whereas the back-arc moves relative to the neighbouring along transform faults (grey shaded regions). Within a short period is old spreading centre and the associate volcanic arc extinct, and a new spreading centre forms closer to the retreating trench.

neighbouring plates at STEP faults (Fig. 1d, Supplementary Fig. 1). At this point, the first back-arc basin is formed: high tear resistance at the STEP faults reduces lateral rollback at the plate interface[31,39], and the resulting trench curvature localizes stresses and ruptures the upper plate. The initial opening of a back-arc basin is facilitated and simplified by this ending of the prescribed transform faults, but subsequent back-arc spreading will involve STEP fault propagation that will form an essential ingredient for back-arc spreading jumps.

Models with oceanic plate widths between 600 and 1000 km show ongoing rollback with back-arc spreading (Fig. 2b). High strain rates at the STEP faults reduce the effective viscosity due to the nonlinear rheology (see Methodology) and allow for tearing

of the slab edge through the lithosphere. Resulting continued trench retreat increases the distance between trench and back-arc spreading centre. Stress and strain rate concentrate at the plate edges and lead to natural plate-like behaviour and decoupling of overriding and neighbouring plates along transform faults. (Fig. 2b). The resulting low frictional resistance of a young and small back-arc plate allows for rollback rates to increase from the initial 3 cm/yr up to 7 cm/yr (Fig. 3a) in this first 5–10 Myrs of back-arc spreading (Fig. 3a).

All deformation in the model, both in the mantle and along plate boundaries at the surface, is entirely driven internally by subduction and localises in the bending downgoing plate, at the trench, at the STEP faults, and at the transform faults between the

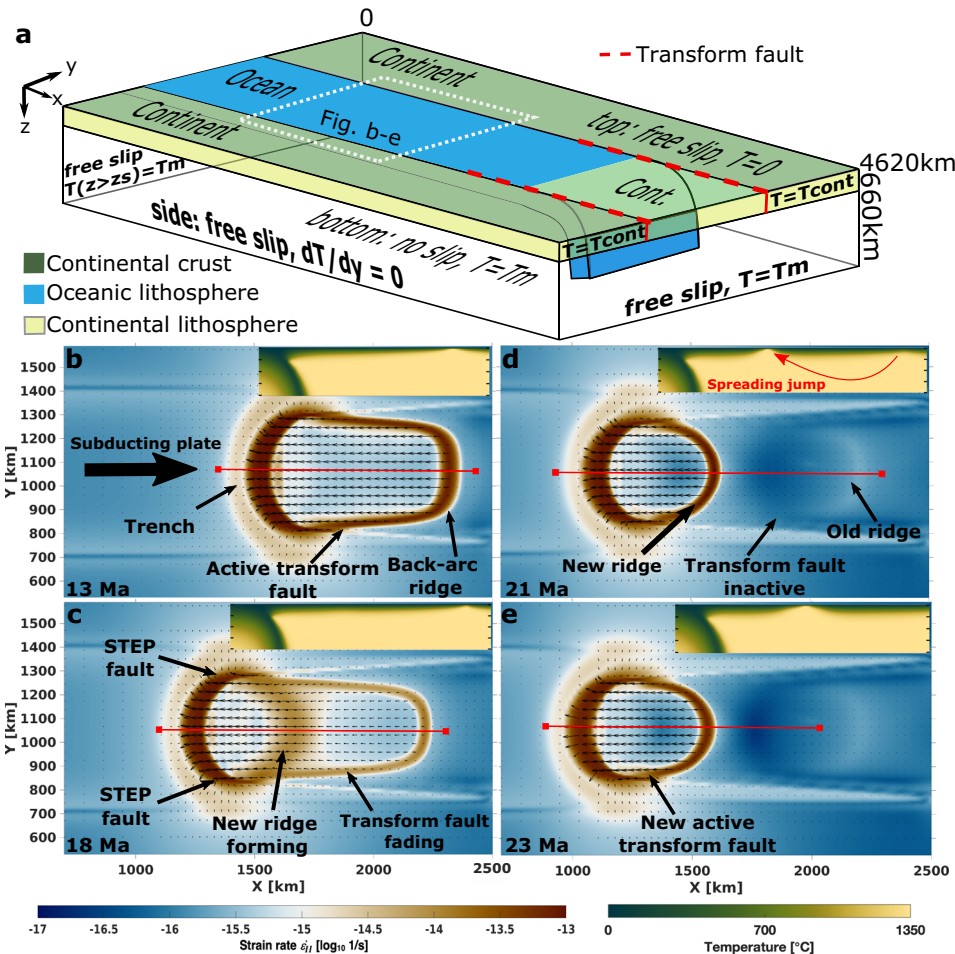

**Fig. 2 Model evolution.** Model setup (**a**) and evolution (**b**–**e**) of the second invariant of the strain rate ($\dot{\varepsilon}_{II} = \sqrt{0.5\dot{\varepsilon}_{ij}^2}$) and temperature for an 800 km wide slab which is used as reference model. Arrows represent absolute motion with reference to the fixed model edge. Model evolution plots show an inset area of panel (**a**) of surface strain rate in the centre of the model domain and the vertical-temperature profile up to 200 km depth along the red line. **a** A narrow slab subducts between two buoyant continents and below an overriding continental plate. The slab is initially placed up to 150 km deep in the mantle and between two prescribed transform faults ending after 660 km in x-direction (dashed red lines) to ensure subduction initiation. **b** After opening of an initial back-arc basin, transform faults form, connecting the trench and spreading centre. **c** At a critical distance, increased coupling between back-arc and neighbouring plates forces the transform faults to become inactive and localization of stresses closer to the trench. **d** A rapid ridge jump occurs. **e** The slab width is decreased and new transform faults are formed.

back-arc spreading centre and neighbouring plates. Of these, only deformation along the transform faults increases with time as the back-arc basin grows, and, as a result, rollback rates reduce (Fig. 3a shows a decrease from 7 to 2–3 cm/yr). Given that the total amount of available energy to overcome resistance against deformation is limited, the transform faults eventually reach a threshold length after which they lock up (Fig. 2c). Ongoing rollback then localises stresses closer to the trench where a new spreading centre fully develops within 2–4 Myr (Fig. 2d, e), while the old spreading centre is abandoned rapidly. This new spreading centre forms where the toroidal flow around the slab edges causes the highest stresses at the base of the overriding plate (see Supplementary Fig. 3)[40]. After the spreading centre jump, the slab width shortens and total deformation at surface is reduced (Fig. 2d), which causes a pulse of increased rollback rate by about 1 cm/yr (Fig. 3a).

The timing of the back-arc spreading jump is governed by the relative strengths of the transform faults and the overriding plate. Wider overriding plates require more energy to break than narrow ones, so that old back-arc spreading centres remain active for longer. Indeed, within the set of models with back-arc spreading jumps, the maximum transform fault length and lifespan of a

back-arc spreading centre both linearly increase with overriding plate width (Fig. 3). A first-order theoretical derivation of supporting this relationship is presented in the Supplementary Discussion. Our conclusion that limited deformation along STEP- and transform faults is crucial for back-arc spreading jumps is supported by model cases that do not develop jumps, and models for a 400 km and a 1500 km wide slab, as well as a model with a weak plate interface along the entire neighbouring plate, are presented in Supplementary Figs. 4–6. Each model has a different reason for the absence of jumps: a 400 km wide slab is unable to overcome tear resistance at STEP faults, i.e. the strain rate induced by the slab pull is too low to weaken the lithosphere and prevents rollback and subduction completely (Fig. 3 and Supplementary Fig. 4). In contrast, the 1500 km wide slab retreats continuously until it reaches the model edge (after >4000 km retreat) without a spreading jump (Supplementary Fig. 5). Although a jump might occur in a longer model domain, such lengthy uninterrupted rollback is unlikely to happen within a typical lifetime of an oceanic plate.

No spreading jumps develop in models with prescribed weak zones (i.e. transform faults) along neighbouring plates (Supplementary Fig. 6). This demonstrates the role of STEP fault tearing

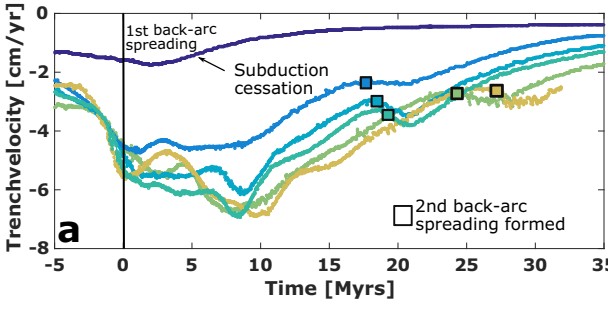

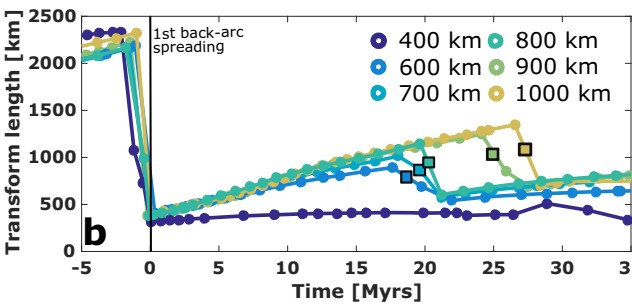

**Fig. 3 Temporal evolution. a** Trench velocities and **b** transform fault lengths for models with various plate widths. Model times are relative to the first back-arc basin opening. Opening of back-arc basins increase rollback rates (i.e. negative trench velocities) temporally. While the overriding plate and transforms grow with ongoing rollback, rollback rates decrease. Spreading jumps are displayed in the suddenly decreased transform length. Subduction ceases in the 400 km model, as well as in the 600 km model after the back-arc spreading centre jump, due to the inability to tear through lithosphere at plates interfaces (see Supplementary Information).

in the development of spreading jumps. With prescribed weaknesses, which are weak and require no energy to form, both the slab and overriding plate can retreat along neighbouring plate effortlessly, making the total transform fault distance between trench and back-arc basin irrelevant. Subducting and overriding plate are basically independent from neighbouring plates and thus spreading centres jumps are not feasible.

## Discussion

Our models show that the ratio of transform fault versus overriding plate strengths governs the wavelength and frequency of back-arc spreading jumps. Narrow plates can only maintain decoupling of back-arc and neighbouring plates along transform faults of finite length. Furthermore, we show why the widest subduction zones are unlikely to undergo any spreading jumps with this mechanism and possibly require external factors or heterogeneities in the plate (e.g., plate re-organisation, lower mantle slab penetration, nearby collision, along-trench variations of trench retreat velocities). Moreover, much wider plates (>2000 km) are likely to have a different trench curvature and velocity along strike due to mantle flow associated with slab rollback[26,41]. To test the applicability of our models, we compare the results to natural back-arcs that have experienced spreading jumps and to those with a single spreading centre (Fig. 1 and 4). Two characteristics are used: (1) The distance between the trench and the centre of the back-arc extension (Fig. 4a, see our Supplementary Tables), which, for our models and natural cases, is equivalent to the transform fault length, and (2) the back-arc rifting/spreading duration (Fig. 4b), which is relatively well constrained in nature by dating of appropriate sedimentary rocks and magnetic seafloor lineations.

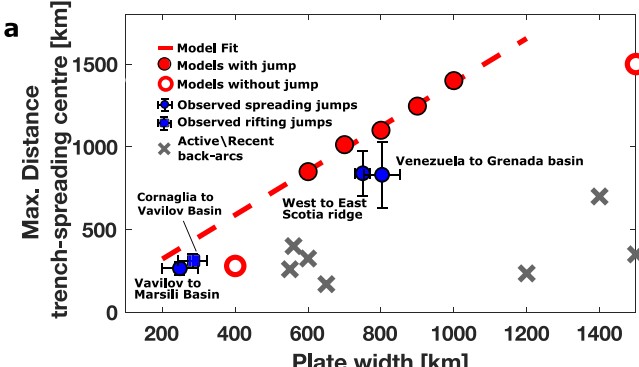

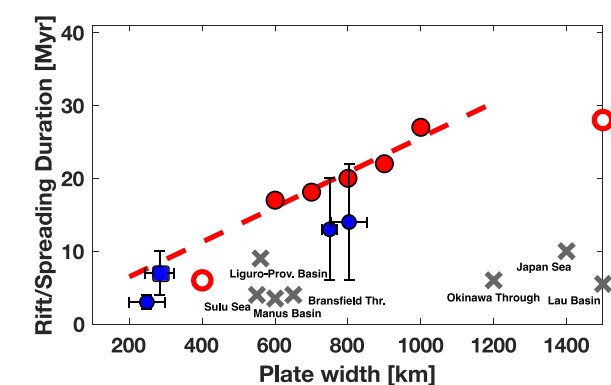

**Fig. 4 Comparison to data.** Comparison of models and observations of back-arc spreading with and without back-arc spreading jumps. Timing in the natural data comprises the entire rifting and spreading duration. **a** The largest distance between trench and back-arc spreading centre. **b** Spreading duration of back-arcs. For models without spreading centre jumps (non-filled red circles), we measure the spreading period of the first back-arc. Models with back-arc spreading jumps form a linear trend, while the ones without spreading jumps plot below this trend. Observational data of back-arc spreading jumps plot close to the regression line, while all other active or recently inactive basins plot below it.

Figure 4 shows that plate widths are linearly correlated to both characteristics in all models with back-arc spreading jumps, while models without jumps (400 km and 1500 km) plot clearly below the regression line. Data points from observations show a primarily short back-arc spreading duration of 3–15 Myrs which also explains short distances between trench and spreading centre (<800 km). Exceptions are subduction systems retreating between continental margins and with spreading jumps, i.e. Tyrrhenian Sea (Calabria), Lesser Antilles and Scotia, which all plot close to the regression line. The regression line thus represents an important threshold: given that we expect spreading jumps to open a new spreading centre and abandon an old spreading centre close to this line, we infer that no active back-arc spreading centre should plot above the line, consistent with natural data.

Our models suggest that the longest subduction zones with back-arc basins, such as the New Hebrides (~1600 km) are not likely to jump, as they would require long-term, uninterrupted rollback of >40 Myrs, assuming our extrapolation is valid at such subduction zone lengths. According to our models, the narrowest plate widths for spreading jumps in the central Mediterranean (<400 km) should not be subducting at all, as lithospheric tearing at STEPs is not feasible. Observations of such subducting narrow plates show, however, that the minimum plate width is lower in nature than in our models. Additional processes reducing tear resistance in nature, and hence also enabling back-arc spreading jumps in narrow subduction systems, could be weaker plate

boundaries[31,39], 'true' brittle deformation at the STEP-faults (see Methodology for rheology), reactivation of old-faults or strain-weakening. Very narrow subduction systems have lower energy dissipation, in which case the exact details of the plate boundary friction become more important, and models could potentially benefit from some of these more complex rheological features[2].

In conclusion, back-arc spreading centres remain active for as long as slip along the lengthening transform faults remains energetically favourable over rifting and rupturing the overriding plate in a new location closer to the subduction zone. Once a threshold length is reached by ongoing rollback, the old spreading centre is locked and a new spreading centre is formed. Given that the available potential energy is proportional to the plate width[9], frictional forces along transform faults might be small compared to the driving buoyancy forces for wide subduction zones, but they are important and non-negligible for narrow subduction zones.

## Data availability

See Supplementary Information for Data used in this study. No new Data was created.

## Code availability

CITCOM is available upon request.

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

## Acknowledgements

This work has been supported by EU FP7 Marie Curie Initial Training Network 'Subi-top', grant agreement No. 674899. JvH acknowledges funding from NERC (grants NE/K010824/1 and NE/M000281/1); V.M. acknowledges support from the Research Council of Norway through its Centres of Excellence funding scheme, Project Number 223272. MBA acknowledges NERC grant NE/H021620/1. We thank Robert Allen for valuable input and Carmen Gaina for fruitful discussions. This work made use of the computational facilities of Hamilton HPC at Durham University.

## Author contributions

N.S. set up the project and the numerical models, and, wrote the manuscript. J.v.H. provided advice on the numerical modelling and contributed to writing the manuscript. M.A. provided the sketches and geological advice. He further provided support in writing the manuscript. V.M. provided the base model on which these models were set up and contributed in writing the manuscript. F.G. provided data and advice on geological implications and contributed in writing the manuscript.

## Competing interests

The authors declare no competing interests.
