## [Peer Review File · Nature Communications]

Episodic back-arc spreading centre jumps controlled by transform fault to overriding plate strength ratioREVIEWER COMMENTS

Reviewer #1 (Remarks to the Author):

Dear editor,

I have read the paper of Schliffke and colleagues with interest. They present a numerical model that shows that in subduction systems with mantle-stationary upper plates bounded by STEPS and dominated by roll-back back-arc spreading centers tend to jump trenchwards, as a result of increase of friction on the upper plate transforms. I am not a modeler, but the experiments look fine to me, and I happily accept that this process may play a role in natural systems.

The paper starts with building a case that the modeled setup is representative for natural systems, and makes that case again at the end of the paper. But contrary to the authors, I find that the natural systems they use (Tyrrhenian, Scotia, Eastern Caribbean) are not similar to the modeled setup as explained in the detailed review below, whereas another (Gibraltar-Algerian basin) is, but does not show a ridge jump. In other words, I don't think that the experiments performed in this paper alone provide an explanation for geologically reconstructed ridge jumps. This point is also admitted by the authors in their discussion, and they point to other potential causes (pre-existing weakness zones, etc) in cases where their model does not apply. But if other causes may play a role there, then how do you assess the importance of your modeled causes?

So rather than making a case that the experiment is based on geological systems, I would suggest that the authors focus their paper on the simple and controlled setup, then discuss how changing that setup would change the result (e.g., would you get the same result if there was no roll-back, but the upper plate moved away from a mantle-stationary trench?), and then discuss to what extent the experiment may aid to explain reconstructed systems. In its present form, I find that the model is too much forced upon the rocks, so to speak, and I think the importance of the model for natural systems is a bit overstated.

I provide my detailed comments below.

All the best,
Douwe van Hinsbergen

I. 22: The Calabrian slab had arguably 2 spreading center jumps (from Gulf de Lion to Vavilov and from Vavilov to Marsili if you consider the small hyperextended portions of the Tyrrhenian Sea spreading centers, and if you consider the Gulf de Lion spreading center, which formed when the subduction zone was still much wider and included most of the Apenninic slabs, also part of the Calabrian slab. The Scotia Sea, however, had at least 7 different spreading centers (Powell, Protector, Dove, Scan, Jane, West Scotia, and East Scotia Basins, arguably also the Central Scotia Basin, and a series of basins with poorly known histories in the northern margin of the basin). So these regions are not illustrative of the point made here.

L. 23: But how about the jump from the Sea of Okhotsk to the Kuril basin, or from the South Fiji Basin to the Lau Basin then? Those are jumps in the back-arcs of very long trenches.

I. 27: from formation at a seafloor spreading center to consumption at a subduction zone

I. 29: Only if the upper plate is not advancing at the same rate as the retreat. Nazca roll-back does not make back-arc basins, but an Andes instead.

I. 30: that may evolve. Basin and Range never went oceanic, or Greece, or Central Anatolia.

I. 31: is the formation, or the location controlled by these features? The formation is simply controlled by the lithosphere being stretched to zero.

I. 32: I don't understand how the formation of a back-arc spreading center can be controlled by the buoyancy of the upper plate or the strength of the lower plate.

I. 32: arcs in models, or in reality?

I. 33: Where would we have an example of this? When buoyant material enters a trench, most models invoke upper plate shortening, and perhaps polarity reversal, not back-arc spreading center formation.

I. 35: How many spreading centers are we talking about here? There is only a handful of oceanic back-arc basins today where we can constrain this, so how do you know this is a representative figure?

I. 38: In both cases, the spreading directions flipped about 90°, are you taking that into account?

I. 39: You seem to infer that the Venezuelan Basin was a back-arc basin of the Lesser Antilles Trench based on recent work of your group. If there was indeed a jump from back-arc spreading in the Venezuelan to the Grenada basins, it must have occurred around the time the Caribbean plate reorganization around 50 Ma, when its relative and absolute motion and its plate boundaries changed dramatically. There is extensive documentation from Colombia, the Belize margin and Yucatan trench, and Cuba that until ~50 Ma, the Caribbean plate moved to the NE relative to NAM. It is possible that the modern northern transform already existed (but the Cayman trough certainly didn't), but the Caribbean plate still shared a northeastward motion together with the Cuban segment north of the Cayman Trench. After 50 Ma, the Caribbean plate became near-mantle stationary and NAM absolute plate motion changed from SW to W, leading to the formation of the Motagua-Cayman plate boundary, and the Lesser Antilles plate boundary, which had been a STEP or a highly oblique subduction zone, became a normal trench. So while the Caribbean example is not straightforward.

Also: you draw three more oceanic basins in the Tyrrhenian Basin than I was not aware of, I only know of Vavilov and Marsili, which seem to be hyperextended portions of the back-arc rather than full spreading centers. Is there any documentation of oceanic crust in those other basins? And if not, why do you include them? Does continental back-arc extension also jump trench-ward? The Aegean example rather shows that upper plate continental extension can be regionally distributed, and active simultaneously over multiple centers (the southward jumping of the onset of extension there seems rather a function of ongoing nappe accretion). So what are these three basins in the Tyrrhenian Basin that you added?

I. 40: What is the large-scale plate reorganization that coincided with the opening of the Lau Basin?

Re: polarity flip: There was a polarity flip to the north of the Lau Basin which narrowed the subduction zone, is that what you mean? But if that was the controlling factor of the ridge jump in the Lau Basin, then something very similar occurred in Scotia: Around 10 Ma, the Antarctica-South America spreading ridge of the Weddell Sea extended towards South Orkney arrived in the trench below the Scotia Sea basins. That led to a narrowing of the subduction zone, a change in upper plate extension direction from ESE to E, and a jump to the East Scotia Basin. So how is Scotia different from Lau?

In addition, you argue that subduction polarity reversal below Taiwan has something to do with a back-arc basin jump in the Parece-Vela basin 2000 km to the east, and 20 million year before the polarity reversal? I don't see how those have much to do with each other? There is also an ongoing polarity reversal of the Algerian basin below North Africa in the last 5 Ma according to some, would that then trigger the jump in the Tyrrhenian Basin?

I. 42: There is no roll-back (relative to the mantle) in the Lesser Antilles trench: the Caribbean plate has been near-mantle stationary since the Eocene. The Scotia Sea only underwent roll-back in the last ~25 Myr when you place the evolution in a mantle frame: all of the pre-25 Ma extension is related to the WNW-ward absolute motion of S America relative to a near-mantle stationary trench, and in total only half of the Scotia Sea extension relates to roll-back, the other half (~1000 km) being upper plate retreat.

In addition: also the Gibraltar subduction zone would be interesting to include. That's also a

narrow subduction zone, and it also retreated rapidly, also had at least one STEP, but there is no ridge jump in the upper plate: all extension appears to have been accommodated in the Algerian Basin (and by extending the Betic-Rif nappe stack see e.g. Mauffret et al 2004; van Hinsbergen et al 2014; 2020; Chertova et al., 2014). Likewise, the Black Sea had no jumps, but did have STEPS. In fact, the western Black Sea had arguably the opposite of what you describe: upper plate extension was first accommodated in the Late Jurassic in the Central Pontide basin, and then jumped in the Early Cretaceous into the upper plate to the Black Sea. The North Fiji basin had a STEP fault and roll-back but not ridge jump that I'm aware of. Does your explanation for ridge jumps also offer explanations for regions that did not behave according to the model?

I. 42: without major plate reorganizations: see earlier remark on the Caribbean around 50: there was a plate reorganization there. Also in Scotia, the extension direction from west Scotia to east Scotia changes by 30° or so (likely associated with a narrowing of the trench and the removal of the E-W portion when the Weddell Sea ridge arrived in the trench).

I. 50: In which case it would be interesting to include also North Fiji, Gibraltar and explain why there were no jumps there.

I. 60-67: Please don't give the conclusions in the introduction, better to do that after showing the results first.

I. 99 The onset of extension in Scotia is estimated around 50 Ma (Livermore for instance), and extension was first WNW-ESE, then E-W after the ridge jump to the East Scotia Sea. But also see the small basins that I mentioned before: all of these have oceanic spreading centers with dated magnetic anomalies, and all of them are much larger than the examples for the Tyrrhenian Sea that you mention, so they are relevant for the discussion...Also, until the arrival of the Weddell Ridge in the trench around the Scotia Sea basins, this subduction system was confined within the South American Plate, not between SAM and ANT.

I. 101: No, the oldest anomalies in the East Scotia Basin are 17 Ma (Larter et al 2003). The arrest of West Scotia Spreading was about 6 Ma (Eagles et al 2005) and the two centers have had considerable temporal overlap. Your statement is inconsistent with marine magnetic anomaly interpretations. So is this really a jump?

I. 104: See earlier remark: at 60 Ma, the Caribbean plate was considerably larger than today, the plate motion was NE-ward, the northern transform of Motagua-Cayman did not exist, the Belize Margin was a STEP, the Lesser Antilles plate boundary was a STEP or highly oblique nascent trench and the plate motion relative to NAM and SAM was 45° different than at 40 Ma. So this case study does not reflect your models setup.

I. 210: Your models show this, but with the chosen boundary conditions.

I. 215: So what are you really saying here? That there are back-arc basin spreading centers, like in the Philippine Sea Plate or in the Lau Basin, that cannot be explained by your model? But where exactly do you then put the boundary between the basins that can and cannot be explained by your model? As discussed, the Caribbean examples you mention occur on either side of a plate reorganization, and the Algerian Basin-Gibraltar system seems to mirror your setup closely, but doesn't have a ridge jump even though you would explain one.

I would recommend to not make the link to reality black and white. No real system behaved like your model: upper plates move, some cases had long slabs reaching the 660 and beyond during the ridge jump while others did not etc etc. So there are some specific jumps where you may argue that friction in upper plate transforms played a role, and other systems where it didn't.

I. 215: I don't think you can 'validate' your model using geological examples, because no geological example mimics your boundary conditions. You could validate it in an analogue lab or so. I think it's more interesting to see where the physical relationship you concluded from your models may aid explaining jumps where no other explanation seems viable, for instance in the Tyrrhenian basin, or in the jumps in the South Scotia Ridge basins (rather than from East to West Scotia, I'd say a link to eastward propagating Weddell Sea ridge subduction is more likely there,

and that 'jump' takes at least 11 Ma)

I. 216: You haven't mentioned rift jumps before and given the widespread evidence that continental crust can widely distribute extension, I think it's not representative to do so. The Croton basin of Calabria also experienced extension before Vavilov did, and is located in the forearc.

I. 220: but see Gibraltar-Algerian basin, where that relationship apparently did not apply.

I. 221: references 41 and 42 are irrelevant here. You have nowhere discussed the Liguro-Provençal basin, nor the South Scotia Ridge. And there are no magnetic sea floor lineations in the Liguro-Provençal basin that I'm aware of, so ref 41 is not appropriate here either. I don't think you need to give a reference for this statement anyway, it's pretty clear how you would date this. Also: either systematically include upper plate rifting (but then include early Basin & Range, Aegean etc), or don't use it (which is what I would recommend since this is yet another can of worms).

I. 226: But the West Scotia Basin was spreading from 27-6 Ma, i.e. 21 Ma, and the East Scotia Basin was for at least 17 Ma and is still active...

I. 228: But Scotia is not between continental margins: the South Scotia Ridge contains continental crust but is part of the upper plate of the subduction system and is bounded to the south by oceanic crust of the Weddell Sea. The modern subduction system is entirely intra-oceanic. And Gibraltar was between continents (Africa and the Balears) since ~16 Ma and rolled back westward over some 800 km until ~8 Ma, but did not behave according to your model, and plots well above your regression line (trench length 150 km, trench-ridge distance 900 km).

I. 234: The Aleutian Basin lithosphere did not form above the Aleutian subduction zone but is trapped Cretaceous crust, see e.g., Vaes et al (2019).

I. 236: So why did the Lau Basin form?

I. 236: what do you mean by plate width? Trench width? But the Calabrian trench in the Pleistocene is much narrower than 400 km right?

I. 242: But if such features play a role in those natural systems that do not behave according to your model, then why would they play no role in the ones that do behave like your model? And if they always play a role, then how do you discern between those causes and the one that you identified?

I. 244: what do you mean? At a subduction zone, there are two plates. Or three when you have a triple junction.

I. 245: This strikes me as valid only for your chosen model setup with a mantle-stationary upper plate and a system entirely driven by roll-back.

Reviewer #2 (Remarks to the Author):

This is a very interesting paper presenting new explanation for the development of ridge jumps in retreating subduction systems. The paper is of broad interest and suits the journal. However, in my opinion the interpretation of results of numerical experiments is partly incorrect and need improvements (see specific comments).

The paper will also gain from presenting simple theory that quantitate the length of the ridge jump from a force balance by using width of the slab and transform/overriding plate strength ratio.
Taras Gerya, Zurich 09.09.2020

Specific comments

Lines 16-18. "Spreading centres jump towards their subduction zone if the distance from trench to spreading centre becomes too long, ~ 1.3 times plate width,..." This length factor should likely depend on the effective frictional resistance along the transform and strength of the backarc lithosphere. The factor should likely be possible to recover from simple force balance calculations.

Fig. 2. Red line in b-e is not explained (vertical cross-section line?). Vertical scale is missing on cross-sections.

Lines 165-167. "Given that the total amount of available energy to overcome frictional resistance against deformation is limited, the transform faults eventually reach a threshold length (~ 1.3 x slab width) after which they lock up (Figure 2c)." The 1.3 factor results from the force balance and depends on the plate and transform strength (that are model parameters). It would be good to present some simple relationship to recover this factor. 1.3 implies that overriding plate (where broken) is roughly 2.6 time stronger than sliding transform of the same length (if active ridge strength is neglected).

Lines 183-189. "Within the set of models with back-arc spreading jumps, the maximum transform fault length and lifespan of a back-arc spreading centre reduce for narrow slabs (Figure 3). The total amount of potential energy from the subducting slab available to drive the slip and deformation along the growing transform faults is, to a large extent, controlled by the slab width: narrow slabs provide less potential energy to enable this deformation, and therefore, the maximum transform fault length is shorter, the upper plate smaller, and, ultimately, causes more frequent spreading centre jumps." I don't think this assumption is correct (see above) but it can be tested by using older (and thus denser) slabs of the same width. I expect that this should not change ridge jump distance that is regulated by transform/overriding plate strength ratio.

Lines 200-202. "No spreading jumps develop in models with prescribed weak zones (i.e. transform faults) along neighbouring plates (Figure S5). This demonstrates the role of STEP fault tearing in the development of spreading jumps." I assume that prescribed transform faults are much weaker than the ones left behind spontaneously propagating STEP faults and thus characteristic ridge jump length factor should be much larger than 1.3. At least this needs to be check.

Lines 210-212. "Our models show that the width of the subducting plate governs the wavelength and frequency of back-arc spreading jumps, since narrow plates can only maintain decoupling of back-arc and neighbouring plates along transform faults of finite length. " Models also suggest that transform/overriding plate strength ratio also control the wavelength. This follows in particular from comparing models with and without weak transforms prescribed along the entire length of the plates (see previous comment).

Lines 229-232. "The regression line thus represents an important threshold: given that we expect spreading jumps to open a new spreading centre and abandon an old rift close to this line, we infer that no active back-arc spreading centre should plot above the line, consistent with natural data." Same as above. The regression line is dependent on the transform/overriding plate strength ratio used in models. For natural systems, this ratio can be thus recovered from the observed length relations for ridge jumps.

lines 243-249. "In conclusion, the episodic spreading centre jumps result from an interaction of all plates at a subduction zone. The finite amount of available driving energy from a single subducting slab limits the length of transform faults that enable back-arc spreading. Once that limited length is reached by ongoing rollback, the old spreading centre is locked and a new spreading centre is formed. Given that the available potential energy is proportional to the plate width, frictional forces along transform faults might be small compared to the driving buoyancy forces for wide subduction zones, but they are important and non-negligible for narrow subduction zones." I don't

think this conclusion is correct. Slab pull controls whether subduction continues or ceases. Ridge jump does need slab pull that is sufficient to break the overriding plate. However, position of the break depends on the transform/overriding plate strength ratio.

Reviewer #3 (Remarks to the Author):

Review of "Episodic back-arc spreading centre jumps controlled by subduction zone length" submitted to Nature Communications by Nicholas Schliffke and co-authors.

In this study, the authors use three-dimensional buoyancy-driven numerical subduction models to investigate back-arc spreading centre jumps self-consistently and to study the effect of slab width (along-trench slab dimension) on their occurrence and location. The authors find that a high resistance on the strike-slip faults bounding the subducting plate must be used in order to successfully model a spreading centre jump. Varying subducting plate width in the range 400-1500 km, the models moreover show a positive correlation between slab width and trench-spreading centre distance. These model results are compared with natural occurrences of spreading centre or rift jumps for four subduction zones, for which the correlation is also positive to the first-order. Implications for natural subduction zones and their spreading centres are that narrow subduction zones produce more closely spaced spreading centres and jumps that are more frequent, whereas wide subduction systems do not cause spreading centre jumps.

1) Significance and novelty of the study:

I find that the investigated question is really interesting and is significant to the community of geoscientists focusing on tectonics and geodynamics in the broad sense. Moreover, the scientific issue addressed by the authors is novel (to the best of my knowledge).

2) Scientific quality/robustness and model limitations:

- The overall methodology uses a valid approach and is well presented, including enough material to enable reproduction of the results. However, I have a comment that needs further reflection from the authors (see below comment).

- The main comment I would like to raise is on the robustness of the models that successfully produce a spreading centre jump. The models that were successful and from which the conclusions are drawn are the ones including self-consistent development of the bounding strike-slip faults and involve a great resistance of these lateral faults. However, these models do not reproduce the same relationship between slab width and trench retreat velocity as the general trend observed in nature (the models display an increase in trench retreat rate with increasing slab width but in nature wide slabs actually produce a slower trench retreat, see e.g. Schellart, 2007, Nature). Therefore, the robustness of these models can be questioned. Do natural subduction zones that present spreading centre jumps show the same relationship as the models? This should be investigated and, if true, it would be an argument in favour of model robustness and it would provide stronger evidence for the conclusions. In the models, trench retreat is strongly resisted by the visco-plastic rheology in the lithospheric portion of the lateral slab edges for those models that do not include a pre-weakened STEP fault along the entire model length. Thus, the models need a relatively high slab pull force to overcome this resistance to subduction and the model with a slab width of 400 km stops since it does not provide enough slab pull. So I am wondering if the resistance at the STEP faults is too high since in nature a 400-km-wide slab can continue to sink into the mantle. The question is: Would reducing the resistance set by the visco-plastic rheology in the lithosphere still allow to reproduce the jumps while changing the relationship between slab width and trench retreat rate? Have such tests been conducted? I do not think my comment should prevent publication but I think that the authors should address it and a minimum

requirement I would have is to discuss it further. Somehow the resistance on these bounding faults is what determines the relationship between slab width and trench-spreading centre distance so it would also be nice to address this question.

- In relation to the above comment, what about spreading centre jumps induced by strong variations in subducting plate velocity? The subduction process and associated velocities are highly time-dependent. So would it be possible that the models with a weak STEP fault over the whole model length actually generates the jump when there is a strong subducting plate velocity reduction followed by an acceleration while the trench continues to retreat (e.g. due to the interaction between the slab and the 660 km discontinuity)?

- In the models, slab penetration into the lower mantle is not simulated, whereas this would change subduction dynamics and kinematics. Could you discuss this limitation?

3) Writing:

- The paper is generally well written and concise. I think a few clarifications are needed in a few places (see line-by-line comments below).

- The text reports that the range of slab widths explored is 400-1000 km, but the authors did run a model with a slab width of 1500 km (figure S4). This should be clarified in the text. Also, in nature the range extends up to ~7000 km so the authors should clarify that wider slabs were not modelled assuming that conclusions obtained with a slab width of 1500 km apply to wider slabs.

- The title could be modified to better reflect the authors' reflection that the increased resistance of the bounding strike-slip faults due to their increased length is the main trigger of the jump.

4) Interpretation of model results:

It is not clear to me whether the STEP faults fade, thus triggering the jump or if the causal relationship is inverse with the jump mostly due to the mantle flow inducing shearing at the base of the overriding plate (or is it a positive feedback mechanism?). Could you clarify your interpretation on the mechanism?

5) Line-by-line comments and suggestions:

L2: "length" should be replaced with "width" to be consistent with the wording used in the paper and with the definition of length and width.

L13: "plate vector changes". This is unclear. What is meant exactly? Change in plate velocity orientation?

L13: "pre-existing weaknesses". This is vague. Where are the weaknesses located?

L18: "friction" may not be an appropriate term. Do you mean "resistance"?

L18-20: "and friction on the boundary transform faults enabling relative motion of back-arc and neighbouring plates becomes too large: the transform faults then lock up, and back-arc spreading ceases". Could this be somewhat clarified or described more concisely? Also, when the resistance on the bounding strike-slip faults is high, this results in a strong decrease in subduction velocity and induced mantle flow. Could this be highlighted better in the abstract?

L73: "high retreat velocities are most likely for thin overriding plates". A reference should be provided to consolidate this statement.

L90-91: To which subduction zone are these back-arc basins related to?

L98: I do not see how "slab stagnation" can create a pulse. To me it could lead to the extinction of a spreading centre, but to then trigger a new spreading centre an acceleration of the subduction process would be needed. Could you develop this part?

L122: It should be "1500 km" for the larger tested slab width.

Figures:

- Fig 2 and Fig S1: The cross-sections should display axes. The red arrow representing the jump process should be labelled.

- Fig 3: It would be useful to show the evolution of the subducting plate velocity so it becomes easier to compare the kinematics with published models that varied slab width.

Ideally a supplementary figure should display the full evolution of velocities (entire model duration e.g. for a reference model) to further clarify that time 0 is when the first spreading centre forms and to provide an overview of the entire model kinematics and dynamics.

- Fig 4: The caption should include what the error bars represent. Also, a coefficient of determination should be given for the regression lines.

References:

A reference that is relevant since it used models that varied slab width is Strak and Schellart (2016). These models could help to discuss the effect of slab width on mantle flow.

Strak, V. and Schellart, W.P., 2016. Control of slab width on subduction-induced upper mantle flow and associated upwellings: Insights from analog models. *Journal of Geophysical Research: Solid Earth*, 121(6), pp.4641-4654.

Comments on "data acquisition" file:

L3: Repetition of "duration".

Table 1: When an average is given the range should also be given.

Comments on "methodology" file:

L41-44: How many particles were used?

How do the surface and maximum yield stresses compare with values used in published subduction models?

Vincent Strak

We thank the editor and all the reviewers for their helpful comments, and give our point-by-point response in red.

REVIEWER COMMENTS

Reviewer #1 (Remarks to the Author):

Dear editor,

I have read the paper of Schliffke and colleagues with interest. They present a numerical model that shows that in subduction systems with mantle-stationary upper plates bounded by STEPS and dominated by roll-back back-arc spreading centers tend to jump trenchwards, as a result of increase of friction on the upper plate transforms. I am not a modeler, but the experiments look fine to me, and I happily accept that this process may play a role in natural systems.

Thanks for this supporting opening statement.

The paper starts with building a case that the modeled setup is representative for natural systems, and makes that case again at the end of the paper. But contrary to the authors, I find that the natural systems they use (Tyrrhenian, Scotia, Eastern Caribbean) are not similar to the modeled setup as explained in the detailed review below, whereas another (Gibraltar-Algerian basin) is, but does not show a ridge jump. In other words, I don't think that the experiments performed in this paper alone provide an explanation for geologically reconstructed ridge jumps. This point is also admitted by the authors in their discussion, and they point to other potential causes (pre-existing weakness zones, etc) in cases where their model does not apply. But if other causes may play a role there, then how do you assess the importance of your modeled causes?

This is a very good point. Our models do not attempt to explain all examples of ridge jumps recorded on Earth, but they do explain how these jumps appear within tectonic settings not complicated by factors such as marked asymmetry and rotation in the system. Our model set-up is a relatively simple plate configuration which nonetheless succeeds in explaining many natural Earth examples. We note where natural exceptions occur, and summarise why they differ from our set-up, but do not make an exhaustive survey of all such cases.

So rather than making a case that the experiment is based on geological systems, I would suggest that the authors focus their paper on the simple and controlled setup, then discuss how changing that setup would change the result (e.g., would you get the same result if there was no roll-back, but the upper plate moved away from a mantle-stationary trench?), and then discuss to what extent the experiment may aid to explain reconstructed systems. In its present form, I find that the model is too much forced upon the rocks, so to speak, and I think the importance of the model for natural systems is a bit overstated.

Yes, we agree with this statement, and have modified the paper to emphasise how our set-up is a simple scenario, that serves as a starting point for understanding natural systems.

I provide my detailed comments below.

All the best,
Douwe van Hinsbergen

l. 22: The Calabrian slab had arguably 2 spreading center jumps (from Gulf de Lion to Vavilov and from Vavilov to Marsili if you consider the small hyperextended portions of the Tyrrhenian Sea spreading centers, and if you consider the Gulf de Lion spreading center, which formed when the subduction zone was still much wider and included most of the Apenninic slabs, also part of the Calabrian slab. The Scotia Sea, however, had at least 7 different spreading centers (Powell, Protector, Dove, Scan, Jane, West Scotia, and East Scotia Basins, arguably also the Central Scotia Basin, and a series of basins with poorly known histories in the northern margin of the basin). So these regions are not illustrative of the point made here.

We see where this point comes from – because the Scotia arc/back-arc system has had more spreading centres overall than the Calabrian system. But the point in the text was that narrower systems like Calabria produce more frequent and closely-spaced jumps, not necessarily a greater overall number of jumps. Figure 1 shows how the back-arc basins of the Calabrian system are much closer together than the Scotia system. It is also a much younger system (14 Ma v 40 Ma). Also, the first five jumps in the Scotia Sea happened during plate reorganisation and before the slab started retreating towards the Atlantic.

L. 23: But how about the jump from the Sea of Okhotsk to the Kuril basin, or from the South Fiji Basin to the Lau Basin then? Those are jumps in the back-arcs of very long trenches.

The nature of the basement in the Sea of Okhotsk is debated, but it is not clear that there was a distinct back-arc spreading centre before oceanic spreading began in the Kuril Basin, ie this is not a case of a spreading centre jump. See Worrall et al (1996, Tectonics).

The modern Lau back-arc basin is young (from 4-5 Ma). It is not clear what is its relation to the South Fiji Basin to its west, because spreading in the latter took place from the Oligocene to Early Miocene, but not later (Herzer et al., 2009). Therefore there was a ~15 million year gap from the end of South Fiji Basin spreading to the onset of the Lau Basin spreading. For this reason we argue that the Lau Basin is not a simple case of ridge jump, regardless of the mechanism.

l. 27: from formation at a seafloor spreading center to consumption at a subduction zone

This has been changed in the text accordingly (line 26).

l. 29: Only if the upper plate is not advancing at the same rate as the retreat. Nazca roll-back does not make back-arc basins, but an Andes instead.

Agreed. Hence the sentence was qualified with the word “most” at the start. The Central Andean Plateau is highly unusual on Earth; there are no other comparable active continental margins.

l. 30: that may evolve. Basin and Range never went oceanic, or Greece, or Central Anatolia.

This has been changed in the text accordingly (line 30).

l. 31: is the formation, or the location controlled by these features? The formation is simply

controlled by the lithosphere being stretched to zero.

It's both. Our models indicate whether back-arc basins form at all, not only their location. 'location' is now added to the text (line 30).

l. 32: I don't understand how the formation of a back-arc spreading center can be controlled by the buoyancy of the upper plate or the strength of the lower plate.

What was meant here was the buoyancy of the subducting plate or the strength of the overriding plate. The original text was somewhat ambiguous, and this is now changed (lines 31-32).

l. 32: arcs in models, or in reality?

Both. The two papers cited at this point are modelling studies (numerical and analogue), but both papers relate their findings to real world examples.

l. 33: Where would we have an example of this? When buoyant material enters a trench, most models invoke upper plate shortening, and perhaps polarity reversal, not back-arc spreading center formation.

The three papers cited at this point cover natural examples where buoyant material has entered a trench and caused rotations (note we do not say whether or not upper plate shortening is involved: we agree that this can happen). The Wallace et al (2009) paper cited contains 23 natural examples where such rotation has happened. See the cited reference 9, Magni et al. (2014): when a variation of buoyancy subducts along the trench, local stress concentrations cause spreading centres to open where trench retreat continues.

l. 35: How many spreading centers are we talking about here? There is only a handful of oceanic back-arc basins today where we can constrain this, so how do you know this is a representative figure?

Again, we have drawn on Wallace et al (2009) for natural examples. Their compilation involves 23 Cenozoic and/or active examples.

l. 38: In both cases, the spreading directions flipped about 90°, are you taking that into account?

We note these re-organisations, and emphasise that our models do not attempt to reproduce this behaviour. We stress that our models work for natural examples such as the ones listed here.

l. 39: You seem to infer that the Venezuelan Basin was a back-arc basin of the Lesser Antilles Trench based on recent work of your group. If there was indeed a jump from back-arc spreading in the Venezuelan to the Grenada basins, it must have occurred around the time the Caribbean plate reorganization around 50 Ma, when its relative and absolute motion and its plate boundaries changed dramatically. There is extensive documentation from Colombia, the Belize margin and Yucatan trench, and Cuba that until ~50 Ma, the Caribbean plate moved to the NE relative to NAM. It is possible that the modern northern transform already existed (but the Cayman trough certainly didn't), but the Caribbean plate still shared a northeastward

motion together with the Cuban segment north of the Cayman Trench. After 50 Ma, the Caribbean plate became near-mantle stationary and NAM absolute plate motion changed from SW to W, leading to the formation of the Motagua-Cayman plate boundary, and the Lesser Antilles plate boundary, which had been a STEP or a highly oblique subduction zone, became a normal trench. So while the Caribbean example is not straightforward.

Yes, we are referring to the jump from the Venezuelan Basin to the Grenada Basin. Although different reconstruction models have been suggested for this region and uncertainties due to paucity of data on the spreading ages exist, we still think that the Lesser Antilles are a fitting example for our model setup and results. In particular, arc volcanism shifts from the Aves ridge (88-59 Ma) to the Lesser Antilles arc (<38 Ma), suggesting that a jump happened between 60 and 40 Ma (Boschmann et al., 2014). Transform faults bounding the overriding plate have been suggested to be active at ~70 Ma and certainly from 50 Ma (e.g., Pindell and Kennan, 2009), which is before and within the range of the time of the jump. We agree that the Caribbean is not a straightforward example due to its complex tectonic setting, however the essence of the scenario, as summarised by Allen et al. (2019) and reproduced in our Fig.1, with a narrow subducting oceanic plate, presence to STEP faults at the edges, and occurrence of a ridge jump during trench retreat, is well-explained by our models.

P.S. The paper referred to here is not by our group, as stated; it is a different “Allen”, and the cited work is not connected to our research group.

Also: you draw three more oceanic basins in the Tyrrhenian Basin than I was not aware of, I only know of Vavilov and Masili, which seem to be hyperextended portions of the back-arc rather than full spreading centers. Is there any documentation of oceanic crust in those other basins? And if not, why do you include them? Does continental back-arc extension also jump trench-ward? The Aegean example rather shows that upper plate continental extension can be regionally distributed, and active simultaneously over multiple centers (the southward jumping of the onset of extension there seems rather a function of ongoing nappe accretion). So what are these three basins in the Tyrrhenian Basin that you added?

The reviewer is right: the three basins are extended continental crust and not always associated with oceanic spreading. We understand that this figure produces confusion and may lead to misunderstanding. Two back arc basins are described in published papers: Vavilov and Masili (Chamot Rooke et al., 1999). In these basins, “real” spreading centres are not always described but instead hyperextension and local subcontinental mantle exhumation, suggesting lithosphere extension and continental break-up (e. g. “real back-arc basins). The Sardinia, Bastia and Cornaglia basins are marked by limited extension and developed at the onset of the Tyrrhenian extension and should have not been put on the figure. Figure 1 and the associated text in the paper have been corrected accordingly. This does, however, not invalidate the Tyrrhenian Basin example, because a jump from Vavilov to Masili occurs (as well as a jump from Liguro Provençal to the Tyrrhenian). It’s a good question whether or not continental extension also jumps trenchward, but not one that our model is set up to answer. We speculate that it may do, based on examples like the Tyrrhenian Sea for example, but the reviewer is right to highlight the Aegean, where it does not. The Aegean is, however, a region of distributed continental extension (wide rifting, Buck 1991) related to high continental geotherm that will most probably never lead to continental break-up (see review in Brun et al. 2016). We modified Figure 1 and associated text, but argue that a discussion whether continental extension undergoes trench-ward jumps is beyond the scope of this manuscript.

l. 40: What is the large-scale plate reorganization that coincided with the opening of the Lau Basin?

Re: polarity flip: There was a polarity flip to the north of the Lau Basin which narrowed the subduction zone, is that what you mean? But if that was the controlling factor of the ridge jump in the Lau Basin, then something very similar occurred in Scotia: Around 10 Ma, the Antarctica-South America spreading ridge of the Weddell Sea extended towards South Orkney arrived in the trench below the Scotia Sea basins. That led to a narrowing of the subduction zone, a change in upper plate extension direction from ESE to E, and a jump to the East Scotia Basin. So how is Scotia different from Lau?

We see two major relevant differences in these two regions: The Lau Basin is very asymmetric – much wider in the north, and narrowing southwards, indicating that its opening relates to the development of strike-slip tectonics at its northern margin, and that considerable rotation is involved. Furthermore, subduction of the New Hebrides Slab started within the last 10 Ma and is now retreating in the opposite direction of the Tonga slab (with the Lau Basin) (Schellart et al., 2006). This significant change of local subduction forces has not occurred in Scotia after 26 Ma, where the Scotia slab remains the only slab in the region. Notwithstanding the points the reviewer makes about similarities between the Scotia and Lau cases, this seems to be an important difference.

In addition, you argue that subduction polarity reversal below Taiwan has something to do with a back-arc basin jump in the Parece-Vela basin 2000 km to the east, and 20 million year before the polarity reversal? I don't see how those have much to do with each other? There is also an ongoing polarity reversal of the Algerian basin below North Africa in the last 5 Ma according to some, would that then trigger the jump in the Tyrrhenian Basin?

We do not make these specific claims about Taiwan. These sentences are phrased as questions, so the answer is i) no and ii) we agree. Polarity flips of the Algerian Basin seem to be a complicated and far-field explanation for the jumps in the Tyrrhenian Basin. We think these examples help make our case: there are natural examples where ridge jumps are caused by far-field tectonic drivers, such as subduction polarity flips, but there are other examples where external drivers are not necessary, because our models demonstrate that the natural development of a subduction system can create ridge jumps and new back-arc basins. It will be a direction for future research to explore the two types, and work out which natural basin fits which scenario. We have offered a starting point in our paper, by summarising major active examples; we do not claim it is exhaustive or conclusive.

l. 42: There is no roll-back (relative to the mantle) in the Lesser Antilles trench: the Caribbean plate has been near-mantle stationary since the Eocene. The Scotia Sea only underwent roll-back in the last ~25 Myr when you place the evolution in a mantle frame: all of the pre-25 Ma extension is related to the WNW-ward absolute motion of S America relative to a near-mantle stationary trench, and in total only half of the Scotia Sea extension relates to roll-back, the other half (~1000 km) being upper plate retreat.

The relative motion in the absolute reference frame plays no significant role in the presented work here. What is important is the relative motion relative to the neighbouring plates. This is clarified further down in the manuscript. We now clarify that we look at rollback compared to fixed neighbouring plates and the upper mantle (lines 40-42).

In addition: also the Gibraltar subduction zone would be interesting to include. That's also a narrow subduction zone, and it also retreated rapidly, also had at least one STEP, but there is no ridge jump in the upper plate: all extension appears to have been accommodated in the Algerian Basin (and by extending the Betic-Rif nappe stack see e.g. Mauffret et al 2004; van Hinsbergen et al 2014; 2020; Chertova et al., 2014).

In the Gibraltar arc, the timing of roll-back, the amount of roll-back and even the vergence of the subduction are still highly debated (see discussion in Chertova et al., 2014). Because the exact link between slab dynamics and upper plate deformation is not clearly established in Gibraltar, we hence prefer to avoid discussing this natural example in the present manuscript. Gueydan et al, (2019) suggest that subduction of continental crust has ceased rollback at around 21 Ma, providing a possible explanation for the absence of "ridge"-jump.

Likewise, the Black Sea had no jumps, but did have STEPS. In fact, the western Black Sea had arguably the opposite of what you describe: upper plate extension was first accommodated in the Late Jurassic in the Central Pontide basin, and then jumped in the Early Cretaceous into the upper plate to the Black Sea.

The Black Sea is a major back-arc basin, but its history is still debated: its age and thick (>10 km) sedimentary cover obscure a lot of the original structure and evolution. According to Okay et al., (2014), extension stopped after subduction of the buoyant Cimmeride zone. Ongoing subduction and uni-directional slab rollback are, however, the basic requirement for our models and the real world example we choose.

The North Fiji basin had a STEP fault and roll-back but not ridge jump that I'm aware of. Does your explanation for ridge jumps also offer explanations for regions that did not behave according to the model?

The model works where the system has two marginal transform (STEP) faults and is reasonably symmetrical, i.e. there are no major rotations. Basins with one or no STEP faults do not conform. The paper makes this distinction, and sets out our aims: "The role of prominent STEP and transform faults in potentially controlling the back-arc spreading jumps has never been constrained". We do not claim to provide a universal mechanism that covers all back-arc basin ridge jumps.

l. 42: without major plate reorganizations: see earlier remark on the Caribbean around 50: there was a plate reorganization there. Also in Scotia, the extension direction from west Scotia to east Scotia changes by 30° or so (likely associated with a narrowing of the trench and the removal of the E-W portion when the Weddell Sea ridge arrived in the trench).

We agree with the reviewer that recorded plate reconstructions in these regions may have played a role in the ridge jumps. But those external triggers are not needed, and, as we will argue later on, unlikely to be the main driver. We adjusted the sentence to reflect this (line 42).

l. 50: In which case it would be interesting to include also North Fiji, Gibraltar and explain why there were no jumps there.

We do not include North Fiji and Gibraltar because both regions have experienced dynamics which differ to our models (see also comments above): North Fiji has experienced plate rotation, while subduction of continental crust in Gibraltar is seen to have ceased subduction. To clarify our choice of comparison, we have added uni-directional rollback (i.e. no significant plate rotation) and ongoing subduction as conditions for our comparison.

l. 60-67: Please don't give the conclusions in the introduction, better to do that after showing the results first.

Okay; these lines are removed.

l. 99 The onset of extension in Scotia is estimated around 50 Ma (Livermore for instance), and extension was first WNW-ESE, then E-W after the ridge jump to the East Scotia Sea. But also see the small basins that I mentioned before: all of these have oceanic spreading centers with dated magnetic anomalies, and all of them are much larger than the examples for the Tyrrhenian Sea that you mention, so they are relevant for the discussion...Also, until the arrival of the Weddell Ridge in the trench around the Scotia Sea basins, this subduction system was confined within the South American Plate, not between SAM and ANT.

In this paper, we discuss back-arc spreading centres that have a similar width compared to the width of the driving subducting plate. We acknowledge there are several minor basins within the Scotia subduction zone (which were mostly formed during plate reorganisation, see comment above), but only the West- and East-Scotia ridges are roughly the same width as the slab. The Calabrian slab is much narrower than the Scotia slab, thus we also study smaller back-arc spreading centres in this region.

l. 101: No, the oldest anomalies in the East Scotia Basin are 17 Ma (Larter et al 2003). The arrest of West Scotia Spreading was about 6 Ma (Eagles et al 2005) and the two centers have had considerable temporal overlap. Your statement is inconsistent with marine magnetic anomaly interpretations. So is this really a jump?

Yes, we acknowledge the possible temporal overlap of spreading of the two ridges in the data acquisition file (and in the large error bars in the data). In the data acquisition file, we have now added a more detailed discussion of the choice of values for Scotia: At 26 Ma the West Scotia Ridge was well established and the East Scotia Ridge has been the dominant spreading centre since 6-8 Ma. Furthermore, we briefly explain how the apparent temporal overlap could be a result of an assumption of the location where the East Scotia Ridge was formed.

l. 104: See earlier remark: at 60 Ma, the Caribbean plate was considerably larger than today, the plate motion was NE-ward, the northern transform of Motagua-Cayman did not exist, the Belize Margin was a STEP, the Lesser Antilles plate boundary was a STEP or highly oblique nascent trench and the plate motion relative to NAM and SAM was 45° different than at 40 Ma. So this case study does not reflect your models setup.

In detail, every natural system will be more complicated than our model set-up. But, like the Scotia and Tyrrhenian Sea cases, it is striking how the internal complexity of each region does not mask the essential point that back-arc ridge jumps do occur in many systems, and that these are explained by our model without the need to reach for an ad hoc explanation in each case.

l. 210: Your models show this, but with the chosen boundary conditions.

Yes! Exactly– the boundary conditions explain some natural examples – notwithstanding their local complexities – and where the models don't work it is because the natural example has something fundamentally different from our set-up, such as major rotation and asymmetry. We have modified this sentence to clarify the setup further (lines 205-219).

l. 215: So what are you really saying here? That there are back-arc basin spreading centers, like in the Philippine Sea Plate or in the Lau Basin, that cannot be explained by your model? But where exactly do you then put the boundary between the basins that can and cannot be explained by your model? As discussed, the Caribbean examples you mention occur on either side of a plate reorganization, and the Algerian Basin-Gibraltar system seems to mirror your setup closely, but doesn't have a ridge jump even though you would explain one. I would recommend to not make the link to reality black and white. No real system behaved like your model: upper plates move, some cases had long slabs reaching the 660 and beyond during the ridge jump while others did not etc etc. So there are some specific jumps where you may argue that friction in upper plate transforms played a role, and other systems where it didn't.

Our models are most applicable in cases with most friction along the plate boundaries. Thus, trenches retreating along two STEP faults are most likely to undergo spreading jumps with this mechanism. The specific point we make is that narrow basins will be expected to produce back-arc ridge jumps on the timescales typical of these systems, and that wider basins (meaning longer parallel to the trench, along strike), take longer to have a ridge jump. For the longest subduction systems, back-arc basins may not form in the 10s of millions of years lifespans of the subduction zone. If a back-arc jump occurs in these systems, it may be triggered by another cause.

But the reviewer is right that some natural systems are too remote from our idealised model setup to draw any useful comparison. E.g., for the Algerian basin-Gibraltar system, the present-day configuration looks like the set-up of our model (e.g. initial conditions of our modelling). However, the configuration of Gibraltar/West Mediterranean at the time of active roll back and back arc extension (e.g. 30 to 20 Ma) was most probably very different, with a wide Apennines slab in continuity from Corsica to the Alboran and with no STEP faults. STEP faults developed progressively at the edge of the systems, as a result of the land-locked basin configuration of the subducting plate and as a result of the progressive northward motion of Africa with respect to Eurasia. The Gibraltar/Alboran area shows no clear back-arc extension since 20 Ma (e.g. Gueydan et al., 2019). STEP faults are supposed to develop after 15 Ma. The Gibraltar system is hence not fitting with our set-up. The Algerian basin is associated with southward rollback of the Apennines slab before collision in Algeria/Tunisia, and is therefore not in direct link with the Gibraltar system.

l. 215: I don't think you can 'validate' your model using geological examples, because no geological example mimics your boundary conditions. You could validate it in an analogue lab or so. I think it's more interesting to see where the physical relationship you concluded from your models may aid explaining jumps where no other explanation seems viable, for instance in the Tyrrhenian basin, or in the jumps in the South Scotia Ridge basins (rather than from East to West Scotia, I'd say a link to eastward propagating Weddell Sea ridge subduction is more likely there, and that 'jump' takes at least 11 Ma)

We agree with the reviewer that ‘validation’ is not the correct term (we changed the sentence, line 213). What was meant here was testing the applicability of the model. Agreed that no natural system will ever mimic our models (or rather, our models do not mimic any natural system). But, as the reviewer points out, the core of our paper is that our models explain the behaviour of natural systems like the Tyrrhenian and Scotia examples, better than any ad hoc mechanism so far.

l. 216: You haven’t mentioned rift jumps before and given the widespread evidence that continental crust can widely distribute extension, I think it’s not representative to do so. The Croton basin of Calabria also experienced extension before Vavilov did, and is located in the forearc.

Note on “rift jumps” deleted here (line 214).

l. 220: but see Gibraltar-Algerian basin, where that relationship apparently did not apply.

See our points on Gibraltar above.

l. 221: references 41 and 42 are irrelevant here. You have nowhere discussed the Liguro-Provençal basin, nor the South Scotia Ridge. And there are no magnetic sea floor lineations in the Liguro-Provençal basin that I’m aware of, so ref 41 is not appropriate here either. I don’t think you need to give a reference for this statement anyway, it’s pretty clear how you would date this. Also: either systematically include upper plate rifting (but then include early Basin & Range, Aegean etc), or don’t use it (which is what I would recommend since this is yet another can of worms).

Fair point. We have deleted these references.

l. 226: But the West Scotia Basin was spreading from 27-6 Ma, i.e. 21 Ma, and the East Scotia Basin was for at least 17 Ma and is still active...

As noted above, the Scotia Sea history is debated.

l. 228: But Scotia is not between continental margins: the South Scotia Ridge contains continental crust but is part of the upper plate of the subduction system and is bounded to the south by oceanic crust of the Weddell Sea. The modern subduction system is entirely intra-oceanic. And Gibraltar was between continents (Africa and the Balears) since ~16 Ma and rolled back westward over some 800 km until ~8 Ma, but did not behave according to your model, and plots well above your regression line (trench length 150 km, trench-ridge distance 900 km).

The nature of the neighbouring plates does not really matter. As long as there are neighbouring plates and friction along the plate boundaries, the jumps should happen. For the Gibraltar system, we are aware of the reviewer’s paper on the region but it is important to note that this geodynamical view is not unequivocally admitted. Other views suggest limited amount of extension since 20 Ma (Jolivet and others) or mainly shortening (Crespo Blanco, Gueydan). So again, the Gibraltar system is not relevant for our study

l. 234: The Aleutian Basin lithosphere did not form above the Aleutian subduction zone but is trapped Cretaceous crust, see e.g., Vaes et al (2019).

We thank the author for pointing this out. Due to this complexity, we decided to remove this example from the text.

l. 236: So why did the Lau Basin form?

Good question – strike-slip tectonics to the north and rotational opening? Opening of such a first back-arc spreading centre is discussed in Magni et al., 2014, but is not relevant for the scope of this study.

l. 236: what do you mean by plate width? Trench width? But the Calabrian trench in the Pleistocene is much narrower than 400 km right?

Yes – but see the next lines, where we discuss why natural examples are narrower than the models.

l. 242: But if such features play a role in those natural systems that do not behave according to your model, then why would they play no role in the ones that do behave like your model? And if they always play a role, then how do you discern between those causes and the one that you identified?

Those features are particularly important for the narrow subduction zones, which have a relatively low energy dissipation, and the details of the implemented rheology are relatively important, whereas for wider, higher energy dissipation subduction zones, those rheological parameter settings become less critical. A detailed assessment of rheological parameters is beyond the scope of this manuscript. But we agree with the reviewer that this point requires further explanation, and therefore we elaborated the text with the following sentence: *“Very narrow subduction systems have lower energy dissipation, in which case the exact details of the plate boundary friction become more important, and models could potentially benefit from some of these more complex rheological features.”*

l. 244: what do you mean? At a subduction zone, there are two plates. Or three when you have a triple junction.

“Single” is deleted here, as it seems to have caused confusion.

l. 245: This strikes me as valid only for your chosen model setup with a mantle-stationary upper plate and a system entirely driven by roll-back.

On the suggestion of reviewer 2, this sentence is now removed completely.

Reviewer #2 (Remarks to the Author):

This is a very interesting paper presenting new explanation for the development of ridge jumps in retreating subduction systems. The paper is of broad interest and suits the journal. However, in my opinion the interpretation of results of numerical experiments is partly incorrect and need improvements (see specific comments).

The paper will also gain from presenting simple theory that quantitate the length of the ridge jump from a force balance by using width of the slab and transform/overriding plate strength ratio.

We sincerely thank the reviewer for the thoughtful comments below, which have significantly improved the manuscript.

Taras Gerya, Zurich 09.09.2020

Specific comments

Lines 16-18. “Spreading centres jump towards their subduction zone if the distance from trench to spreading centre becomes too long, ~ 1.3 times plate width,…” This length factor should likely depend on the effective frictional resistance along the transform and strength of the backarc lithosphere. The factor should likely be possible to recover from simple force balance calculations.

We have deleted the specific number and added the force balance in the discussion. See later comments for further details.

Fig. 2. Red line in b-e is not explained (vertical cross-section line?). Vertical scale is missing on cross-sections.

The red lines are explained in the caption. We have added the vertical scale in the cross-section plots.

Lines 165-167. “Given that the total amount of available energy to overcome frictional resistance against deformation is limited, the transform faults eventually reach a threshold length (~ 1.3 x slab width) after which they lock up (Figure 2c).” The 1.3 factor results from the force balance and depends on the plate and transform strength (that are model parameters). It would be good to present some simple relationship to recover this factor. 1.3 implies that overriding plate (where broken) is roughly 2.6 time stronger than sliding transform of the same length (if active ridge strength is neglected).

We thank the reviewer for this insightful remark. Using this, we are now able to provide an improved and more quantitative explanation of the force balance, which indeed includes the strength of the overriding plate and the transform faults. A theoretical derivation of the linear relationship between these is now provided in the Supplementary Material. In this derivation, the factor 1.3 will depend on the chosen rheological parameters, and is therefore not a universal result. We therefore decided not to mention this factor anymore in the new version of the manuscript.

Lines 183-189. “Within the set of models with back-arc spreading jumps, the maximum transform fault length and lifespan of a back-arc spreading centre reduce for narrow slabs (Figure 3). The total amount of potential energy from the subducting slab available to drive the slip and deformation along the growing transform faults is, to a large extent, controlled by the slab width⁹: narrow slabs provide less potential energy to enable this deformation, and therefore, the maximum transform fault length is shorter, the upper plate smaller, and, ultimately, causes more frequent spreading centre jumps.” I don’t think this assumption is

correct (see above) but it can be tested by using older (and thus denser) slabs of the same width. I expect that this should not change ridge jump distance that is regulated by transform/overriding plate strength ratio.

We agree with the reviewer: the linear relationship between the transform fault length and slab width is because slab width is also the width of the overriding plate in these models, and, as explained with the reviewer's previous point, this is now addressed in the main text and supplementary material. The text above has been replaced with (lines 179-184): *“The timing of the back-arc spreading jump is governed by the relative strengths of the transform faults and the overriding plate. Wider overriding plates require more energy to break than narrow ones, so that old back-arc spreading centres remain active for longer. Indeed, within the set of models with back-arc spreading jumps, the maximum transform fault length and lifespan of a back-arc spreading centre both linearly increase with overriding plate width (Figure 3). A first-order theoretical derivation of this relationship is presented in the supplementary material.”*

Lines 200-202. “No spreading jumps develop in models with prescribed weak zones (i.e. transform faults) along neighbouring plates (Figure S5). This demonstrates the role of STEP fault tearing in the development of spreading jumps.” I assume that prescribed transform faults are much weaker than the ones left behind spontaneously propagating STEP faults and thus characteristic ridge jump length factor should be much larger than 1.3. At least this needs to be checked.

Yes, this is correct: weak pre-existing transform faults can remain active for much longer. This is now demonstrated quantitatively in the Supplementary Material. In addition, the following sentence in the manuscript was modified (lines 198-200): *“With prescribed weaknesses, which are weak and require no energy to form, both the slab and overriding plate can retreat along neighbouring plate effortlessly, making the total transform fault distance between trench and back-arc basin irrelevant.”*

Lines 210-212. “Our models show that the width of the subducting plate governs the wavelength and frequency of back-arc spreading jumps, since narrow plates can only maintain decoupling of back-arc and neighbouring plates along transform faults of finite length. “ Models also suggest that transform/overriding plate strength ratio also control the wavelength. This follows in particular from comparing models with and without weak transforms prescribed along the entire length of the plates (see previous comment).

This relates to the previous comments, and, again, we agree with the reviewer. This is now explicitly changed in the text (lines 205-206: “Our models show that the ratio of transform fault versus overriding plate strengths governs the wavelength and frequency of back-arc spreading jumps”) and is further addressed in the Supplementary Material.

Lines 229-232. “The regression line thus represents an important threshold: given that we expect spreading jumps to open a new spreading centre and abandon an old rift close to this line, we infer that no active back-arc spreading centre should plot above the line, consistent with natural data.” Same as above. The regression line is dependent on the transform/overriding plate strength ratio used in models. For natural systems, this ratio can be thus recovered from the observed length relations for ridge jumps.

Again, we agree with the reviewer, and the derivation of the regression line is now derived (in the Supplementary Material) using the energy balance between the transform fault strength and overriding plate strength.

lines 243-249. “In conclusion, the episodic spreading centre jumps result from an interaction of all plates at a subduction zone. The finite amount of available driving energy from a single subducting slab limits the length of transform faults that enable back-arc spreading. Once that limited length is reached by ongoing rollback, the old spreading centre is locked and a new spreading centre is formed. Given that the available potential energy is proportional to the plate width, frictional forces along transform faults might be small compared to the driving buoyancy forces for wide subduction zones, but they are important and non-negligible for narrow subduction zones.” I don't think this conclusion is correct. Slab pull controls whether subduction continues or ceases. Ridge jump does need slab pull that is sufficient to break the overriding plate. However, position of the break depends on the transform/overriding plate strength ratio.

In the light of the above changes, this concluding remark has now also changed to (lines 243-246): *“In conclusion, back-arc spreading centres remain active for as long as slip along the lengthening transform faults remains energetically favourable over rifting the overriding plate in a new location closer to the subduction zone. Once a threshold length is reached by ongoing rollback, the old spreading centre is locked and a new spreading centre is formed.”*.

Reviewer #3 (Remarks to the Author):

Review of "Episodic back-arc spreading centre jumps controlled by subduction zone length" submitted to Nature Communications by Nicholas Schliffke and co-autors.

In this study, the authors use three-dimensional buoyancy-driven numerical subduction models to investigate back-arc spreading centre jumps self-consistently and to study the effect of slab width (along-trench slab dimension) on their occurrence and location. The authors find that a high resistance on the strike-slip faults bounding the subducting plate must be used in order to successfully model a spreading centre jump. Varying subducting plate width in the range 400-1500 km, the models moreover show a positive correlation between slab width and trench-spreading centre distance. These model results are compared with natural occurrences of spreading centre or rift jumps for four subduction zones, for which the correlation is also positive to the first-order. Implications for natural subduction zones and their spreading centres are that narrow subduction zones produce more closely spaced spreading centres and jumps that are more frequent, whereas wide subduction systems do not cause spreading centre jumps.

1) Significance and novelty of the study:

I find that the investigated question is really interesting and is significant to the community of geoscientists focusing on tectonics and geodynamics in the broad sense. Moreover, the scientific issue addressed by the authors is novel (to the best of my knowledge).

Thank you for this supporting statement.

2) Scientific quality/robustness and model limitations:

- The overall methodology uses a valid approach and is well presented, including enough

material to enable reproduction of the results. However, I have a comment that needs further reflection from the authors (see below comment).

Thank you for this supporting statement too.

- The main comment I would like to raise is on the robustness of the models that successfully produce a spreading centre jump. The models that were successful and from which the conclusions are drawn are the ones including self-consistent development of the bounding strike-slip faults and involve a great resistance of these lateral faults. However, these models do not reproduce the same relationship between slab width and trench retreat velocity as the general trend observed in nature (the models display an increase in trench retreat rate with increasing slab width but in nature wide slabs actually produce a slower trench retreat, see e.g. Schellart, 2007, Nature). Therefore, the robustness of these models can be questioned. Do natural subduction zones that present spreading centre jumps show the same relationship as the models? This should be investigated and, if true, it would be an argument in favour of model robustness and it would provide stronger evidence for the conclusions.

The relationship between trench retreat velocity and slab width from Schellart et al. (2007) is valid for a larger range of slab widths than the one we explore in our study. Schellart et al. make a distinction between narrow slabs (width $W \leq 1500$ km), intermediate slabs ($W = 2000$ - 3000 km), and wide slabs ($W \geq 4000$ km), showing that they behave differently in terms of trench migration; for narrow slabs, trench retreat is largest in the centre forming a curved and concave trench, whereas intermediate and wide slabs retreat slower in the centre and faster at the edges resulting in either rectilinear or convex shapes of the trench. In our work, we study only the 'narrow slab' group (according to the definition of Schellart et al.), as we model slabs with $W \leq 1500$ km. And indeed, in our models trench retreat is always largest in the centre resulting in a curved and concave shape, in agreement with observations from Schellart et al.. Therefore, we think that our models are robust as they manage to reproduce the behaviour of narrow slabs that is found in nature and observed in Schellart et al.. We added this point in the discussion (lines 211-213)

In the models, trench retreat is strongly resisted by the visco-plastic rheology in the lithospheric portion of the lateral slab edges for those models that do not include a pre-weakened STEP fault along the entire model length. Thus, the models need a relatively high slab pull force to overcome this resistance to subduction and the model with a slab width of 400 km stops since it does not provide enough slab pull. So I am wondering if the resistance at the STEP faults is too high since in nature a 400-km-wide slab can continue to sink into the mantle.

Yes, we agree with this point. In the discussion section (lines 239-242) , we provide possible reasons why narrower subduction zones are subducting, unlike in our models. We further elaborated this discussion to make this point even clearer (see next point).

The question is: Would reducing the resistance set by the visco-plastic rheology in the lithosphere still allow to reproduce the jumps while changing the relationship between slab width and trench retreat rate? Have such tests been conducted? I do not think my comment should prevent publication but I think that the authors should address it and a minimum requirement I would have is to discuss it further. Somehow the resistance on these bounding faults is what determines the relationship between slab width and trench-spreading centre distance so it would also be nice to address this question.

The reviewer is right, and we tried to elaborate on this discussion further to explain this better. We also added a further disclaimer to the text: *“Very narrow subduction systems have lower energy dissipation, in which case the exact details of the plate boundary friction become more important, and models could potentially benefit from some of these more complex rheological features.”*

- In relation to the above comment, what about spreading centre jumps induced by strong variations in subducting plate velocity? The subduction process and associated velocities are highly time-dependent. So would it be possible that the models with a weak STEP fault over the whole model length actually generates the jump when there is a strong subducting plate velocity reduction followed by an acceleration while the trench continues to retreat (e.g. due to the interaction between the slab and the 660 km discontinuity)?

We agree with the reviewer that strong variations in subducting plate velocity might be important for ridge jumps. Our models allow for subduction velocity variations within a homogenous subduction system. In fact, we do observe a change in subduction velocity due to the interaction between slab and 660 km discontinuity, but, in our models, that is not enough to cause a ridge jump. More complex settings including features such as heterogeneous plates, along trench collisions, trench rotation, interaction with plumes, and regional plate reorganization might produce large velocity variations and have important effects on the behaviour spreading jumps. This is however beyond the scope of our study, which aims at showing how spreading centres jumps are controlled by transform faults and overriding plate strength in homogenous and narrow subduction systems.

- In the models, slab penetration into the lower mantle is not simulated, whereas this would change subduction dynamics and kinematics. Could you discuss this limitation?

While we agree that this may trigger back-arc ridge jumps, just like other external triggers such as plate re-organisations, mantle plumes or nearby continental collision, the key point of our presented model is that it does not need such triggers. The 2nd paragraph of the ‘comparison to natural systems’ discusses the potential role of external triggers, and we added lower mantle slab penetration as additional possibility (line 210).

3) Writing:

- The paper is generally well written and concise. I think a few clarifications are needed in a few places (see line-by-line comments below).

Thank you for this supporting statement.

- The text reports that the range of slab widths explored is 400-1000 km, but the authors did run a model with a slab width of 1500 km (figure S4). This should be clarified in the text. Also, in nature the range extends up to ~7000 km so the authors should clarify that wider slabs were not modelled assuming that conclusions obtained with a slab width of 1500 km apply to wider slabs.

We have changed the text to also include the model with a 1500 km wide slab in the description of the explored parameter space. The reviewer is right that in nature the range of slab widths is larger than what we explore here. This is indeed because any slabs wider than 1500 km don’t exhibit spontaneous back-arc ridge jumps, both in nature and in our models,

and are therefore beyond the scope of this manuscript. We explain this in the discussion (lines 208-213): “*Furthermore, we show why the widest subduction zones are unlikely to undergo any spreading jumps with this mechanism and possibly require external factors or heterogeneities in the plate (e.g., plate re-organisation, nearby collision, along-trench variations of trench retreat velocities). Moreover, much wider plates (>2000 km) are likely to have a different trench curvature and velocity along strike due to mantle flow associated with slab rollback.*”

- The title could be modified to better reflect the authors' reflection that the increased resistance of the bounding strike-slip faults due to their increased length is the main trigger of the jump.

We agree with this suggestion. Also with respect to the previous reviewers comments, we have changed the title to “*Episodic back-arc spreading centre jumps controlled by transform fault to overriding plate strength ratio*”.

4) Interpretation of model results:

It is not clear to me whether the STEP faults fade, thus triggering the jump or if the causal relationship is inverse with the jump mostly due to the mantle flow inducing shearing at the base of the overriding plate (or is it a positive feedback mechanism?). Could you clarify your interpretation on the mechanism?

After suggestions from Reviewer 2, we have now changed the line of argument and explain that it is the ratio between the strength of the STEP faults and the overriding plate that control the occurrence of the jump. We clarify this point throughout the manuscript.

5) Line-by-line comments and suggestions:

L2: "length" should be replaced with "width" to be consistent with the wording used in the paper and with the definition of length and width.

We have changed the title and deleted the word “width”.

L13: "plate vector changes". This is unclear. What is meant exactly? Change in plate velocity orientation?

We have changed the text to the reviews suggestion.

L13: "pre-existing weaknesses". This is vague. Where are the weaknesses located?

L18: "friction" may not be an appropriate term. Do you mean "resistance"?

We agree. We have changed the wording throughout the manuscript.

L18-20: "and friction on the boundary transform faults enabling relative motion of back-arc and neighbouring plates becomes too large: the transform faults then lock up, and back-arc spreading ceases". Could this be somewhat clarified or described more concisely? Also, when the resistance on the bounding strike-slip faults is high, this results in a strong decrease in subduction velocity and induced mantle flow. Could this be highlighted better in the abstract?

We rephrase this part also in the light of the new line of argument. We however do not discuss in the abstract the decrease in subduction velocity and induced mantle flow, since it is not key to our main point and wording in the abstract is very limited..

L73: "high retreat velocities are most likely for thin overriding plates". A reference should be provided to consolidate this statement.

We added references Holt et al., 2015; Hertgen et al., 2020

L90-91: To which subduction zone are these back-arc basins related to?

These back-arc basins belong to the Celebes Sea and New Caledonia subduction zones. We now specify it in the manuscript (lines 85-86).

L98: I do not see how "slab stagnation" can create a pulse. To me it could lead to the extinction of a spreading centre, but to then trigger a new spreading centre an acceleration of the subduction process would be needed. Could you develop this part?

We have changed the explanation to "accelerated slab retreat after slab tip stagnation on top of the lower mantle".

L122: It should be "1500 km" for the larger tested slab width.

Thanks. This has now been changed throughout the manuscript.

Figures:

- Fig 2 and Fig S1: The cross-sections should display axes. The red arrow representing the jump process should be labelled.

We have changed both figures accordingly.

- Fig 3: It would be useful to show the evolution of the subducting plate velocity so it becomes easier to compare the kinematics with published models that varied slab width. Ideally a supplementary figure should display the full evolution of velocities (entire model duration e.g. for a reference model) to further clarify that time 0 is when the first spreading centre forms and to provide an overview of the entire model kinematics and dynamics.

The subducting plate velocity is near zero, as we do not model a mid-ocean-ridge for the subducting plate. For this reason, we do not add the subducting plate velocity. We have added a figure presenting the velocities for the entire reference model (Figure S2).

- Fig 4: The caption should include what the error bars represent. Also, a coefficient of determination should be given for the regression lines.

The coefficient of the regression line is now explained within text (see also previous reviewers comments).

References:

A reference that is relevant since it used models that varied slab width is Strak and Schellart (2016). These models could help to discuss the effect of slab width on mantle flow.

Strak, V. and Schellart, W.P., 2016. Control of slab width on subduction-induced upper mantle flow and associated upwellings: Insights from analog models. *Journal of Geophysical Research: Solid Earth*, 121(6), pp.4641-4654.

We added this reference in line 213

Comments on "data acquisition" file:
L3: Repetition of "duration".

We have changed the document accordingly.

Table 1: When an average is given the range should also be given.

The three averaged values in Table 1 "Active/ceased back-arc spreading" have a range. All other numbers are not averaged.

Comments on "methodology" file:
L41-44: How many particles were used?

The number has been added.

How do the surface and maximum yield stresses compare with values used in published subduction models?

These values have been chosen according to Magni et al., (2014).

Vincent Strak

REVIEWER COMMENTS

Reviewer #1 (Remarks to the Author):

Dear editor,

I have read the rebuttal and discussion of my review by Schliffke and colleagues, and the revised paper. Much of my review, and the rebuttal to that, discussed to what extent the modeled setup may or may not explain natural examples of ridge jumps in back-arc basins, and I have a few comments below on these discussions. But after reading the paper again, I keep wondering what I really learned from it. The paper essentially makes one point: in a 3D numerical model, ridge jumps can be made by tuning the strength of the upper plate and the friction of transform faults. Whether subduction really plays a critical role here, I'm not sure – since the previous review, I have found that a Cretaceous ridge jump in the South Atlantic, between two major transform faults to the east of the Falkland Plateau (Malvinas Plate, see e.g. Marks & Stock, *Mar Geophys Res* 2001) may be the closest example to the modeled setup and this ridge jump may be well explained by the length of the transforms. Then again, the basins between transforms in the South Scotia Ridge were mostly simultaneously active, so apparently the mechanism did not work there. The generic point of this paper is in principle logical. Whether or not this explains back-arc basin jumps, I'm not so sure. As pointed out by the authors, there are plenty of back-arc basin spreading centers that either did not jump, or jumped after a time delay, or jumped during rotation, or jumped without transforms present (and I think two of the three systems that the authors discuss fall in these categories) and these may have different triggers.

I still have a few problems with the application of the modelled setup to the Caribbean and Scotia cases and I find the responses of the authors not convincing. The authors write in their response that: "Our models do not attempt to explain all examples of ridge jumps recorded on Earth, but they do explain how these jumps appear within tectonic settings not complicated by factors such as marked asymmetry and rotation in the system. Our model setup is a relatively simple plate configuration which nonetheless succeeds in explaining many natural Earth examples. We note where natural exceptions occur, and summarise why they differ from our set-up, but do not make an exhaustive survey of all such cases". But the Caribbean case is most certainly associated with rotation in the system: the Venezuela basin opening, if dated correctly, prior to a reorganization of plate boundaries and plate motion vectors, after which the Grenada basin formed. Moreover, the Grenada basin does not, as in the model, continue from STEP to STEP, but is only located in the southern 2/3 of the plate. The northern 1/3, from the latitude of Guadeloupe to the north actually underwent shortening from arc to trench during opening of the Grenada basin (Philippon et al, *PLOS One* 2020). So the Grenada basin was bounded only on one side by a STEP, comparable to the Lau basin that the authors argue cannot be explained by their model.

Likewise, there is an angle of 20° between the extension directions of the West Scotia Sea and East Scotia Sea. In my review I explained how this resulted from a reorganization of the trench due to ridge arrival, and the reconstruction on which I based those remarks was recently published in van de Lagemaat et al (*Earth-Sci Rev* 2021). Moreover, there is the issue that the West and East Scotia ridges were simultaneously active for at least 11 Ma: the authors argue against this now in a table in their Supplementary Data section. They write: "We here point out that the interpretation of spreading times in the East Scotia Ridge with observed chrons (e.g. Figure 4 in Eagles & Jokat (2014)) assume the East Scotia Ridge to have been formed close of the trench. The chrons can be interpreted differently, if we assume the East Scotia Ridge was formed within the basin created by the West Scotia Ridge: This leads to the chrons 3A (6-8 Ma) to be the only chron to be associated to the East Scotia ridge. The chrons interpreted as 5 and 5C in Eagles & Jokat (2014) would have been created by the West Scotia Ridge, not the East Scotia Ridge." This interpretation is undefendable: (i) the West and East Scotia Basins are separated by the Central Scotia Basin that has E-W trending magnetic anomalies, and even though the origin of these anomalies and this crust is debated (either early, trench-parallel upper plate extension, or trapped pre-subduction initiation South American lithosphere) there is a long-standing consensus that the crust of the Central Scotia Basin is older than both the West and East Scotia Basins. (ii) the anomalies in the West Scotia basin have a different orientation than in the East Scotia Basin; (iii) all anomalies older than chrons 3A are accounted for in the West Scotia Basin on either side of the ridge. So the anomalies in the East Scotia Basin were not created in the West Scotia Basin, but to the east of the Central Scotia Basin. And there is no marked change in spreading direction or rate

between the oldest anomaly (C5Cn.3n.o) and the present-day, see Larter et al 2003; Fig 4 in van de Lagemaat et al 2021), so the argument that the East Scotia basin became the 'dominant' spreading center is not defensible: first there were two spreading centers, and then there was one.

Finally, in the Calabrian case, the hyperextended, magma-poor Marsili ridges that is more trenchward than the somewhat older Vavilov ridge are interpreted here as a 'ridge jump'. But is this a ridge jump? These hyperextended portions formed in an upper plate that consists of nappes and continental crust, and this crust has been extended since the Miocene along major detachments, from the Sardinia margin all the way to the Crotone basin. At some stage, there was no continent left to stretch, and instead mantle rocks were exhumed, but the first arrival of those mantle rocks at the surface and the associated melting, which is a bit younger in Marsili than in Vavilov, do not mark the onset of upper plate extension, nor an age progression of extension towards the trench. The extension in the Crotone basin in Calabria, or elsewhere in the Calabrian nappe stack, long predates the formation of Vavilov and Marsili. The model of a ridge jump in an oceanic upper plate does not compare to the Tyrrhenian basin.

In their response, the authors write: "In detail, every natural system will be more complicated than our model set-up. But, like the Scotia and Tyrrhenian Sea cases, it is striking how the internal complexity of each region does not mask the essential point that back-arc ridge jumps do occur in many systems, and that these are explained by our model without the need to reach for an ad hoc explanation in each case." Perhaps there is no need to 'reach for an ad hoc explanation', but in all cases discussed by the authors, the kinematic history simply shows that the relocation of the accommodation of extension coincided with reorganizations of the plate boundary (Caribbean, Scotia), or that the 'jump' is not a jump, but simply a different moment of exhumation of mantle rock after a prolonged, distributed extension history. These reconstructions were not made to find an ad hoc explanation for ridge jumps, the explanations simply follow from the sequence of events.

So what does this modeling study really solve? What is the importance of the conclusion that in some very specific cases, ridges may jump because transforms become too long?

The authors formulate this importance as: "Our models explain why narrow subducting plates (e.g. Calabrian slab), have more frequent and closely-spaced spreading jumps than wider subduction zones (e.g. Scotia). It also explains why wide back-arc basins undergo no spreading centre jumps in their life cycle." But the back-arc basins that are discussed are few, and have one, maybe two ridge jumps, and 'frequency', 'closely-spaced', 'narrow' and 'wider' are not quantified in this paper, and I don't think they can be quantified or substantiated. Second, even if the conclusion is valid, what is so important about this?

I am sorry to be so critical, and I am well aware that reality is always more complex than model setups. But perhaps the authors can explain what new opportunities are created by understanding why ridges between transforms above subduction zones jump, and how that will help us to advance predictability of geodynamic systems?

Utrecht, May 31, 2021
Douwe van Hinsbergen

Reviewer #2 (Remarks to the Author):

The Authors properly revised the paper that I can now recommend for publication.
Taras Gerya, 04.06.2021, Zurich

Reviewer #3 (Remarks to the Author):

Review of revised manuscript "Episodic back-arc spreading centre jumps controlled by transform fault to overriding plate strength ratio" submitted to Nature Communications by Nicholas Schliffke and co-authors.

In my opinion, the authors have improved the manuscript sufficiently according to the reviewer comments to warrant its publication in the journal.

Kind regards,

Vincent Strak

We thank all three reviewers for their comments on the revised version of the paper, and are very pleased that reviewers #2 and #3 concluded that this version of the paper is ready for publication. We therefore focus our responses this time on the comments of reviewer #1. His detailed remarks are very helpful; we agree with his comments about the Scotia system and have revised accordingly. We note the remarks about the Caribbean, but do not need to modify the paper here. In neither case do the local complexities invalidate our models. We disagree with his assessment of the Calabrian system. It is arguably a strength of our models that they still explain the first order behaviour of these and other natural examples, despite the inevitable complexities of nature.

Reviewer #1 (Remarks to the Author):

Dear editor,

I have read the rebuttal and discussion of my review by Schliffke and colleagues, and the revised paper. Much of my review, and the rebuttal to that, discussed to what extent the modeled setup may or may not explain natural examples of ridge jumps in back-arc basins, and I have a few comments below on these discussions. But after reading the paper again, I keep wondering what I really learned from it. The paper essentially makes one point: in a 3D numerical model, ridge jumps can be made by tuning the strength of the upper plate and the friction of transform faults. Whether subduction really plays a critical role here, I'm not sure – since the previous review, I have found that a Cretaceous ridge jump in the South Atlantic, between two major transform faults to the east of the Falkland Plateau (Malvinas Plate, see e.g. Marks & Stock, Mar Geophys Res 2001) may be the closest example to the modeled setup and this ridge jump may be well explained by the length of the transforms. Then again, the basins between transforms in the South Scotia Ridge were mostly simultaneously active, so apparently the mechanism did not work there.

Whilst we are interested that the reviewer has provided another example of a ridge jump, we note that in the case of the Malvinas Plate it is not in a back-arc setting and it is not related to any subduction zone. Therefore, although it would be interesting to apply our conceptual model of ridge jump related to the ratio between the strength of transform faults and the strength of the plate, the example suggested by the reviewer does not fit in with our study.

The generic point of this paper is in principle logical. Whether or not this explains back-arc basin jumps, I'm not so sure. As pointed out by the authors, there are plenty of back-arc basin spreading centers that either did not jump, or jumped after a time delay, or jumped during rotation, or jumped without transforms present (and I think two of the three systems that the authors discuss fall in these categories) and these may have different triggers.

We have not only pointed out the difference between the behaviour between different back-arc spreading centres, but provided a physical mechanism why these differences occur – which is a major advance in this study. In particular, our models explain why wider basins do not evolve with spreading centre jumps. We do not claim it is the only control on ridge jumps, nor does our model explain other behaviour and complications in the natural systems, such as plate rotations. To make this clearer in the manuscript, we explicitly mention this now in the abstract (lines 20-21).

I still have a few problems with the application of the modelled setup to the Caribbean and Scotia cases and I find the responses of the authors not convincing. The authors write in their response that: "Our models do not attempt to explain all examples of ridge jumps recorded on Earth, but they do explain how these jumps appear within tectonic settings not complicated by factors such as marked asymmetry and rotation in the system. Our model setup is a relatively simple plate configuration which nonetheless succeeds in explaining

many natural Earth examples. We note where natural exceptions occur, and summarise why they differ from our set-up, but do not make an exhaustive survey of all such cases". But the Caribbean case is most certainly associated with rotation in the system: the Venezuela basin opening, if dated correctly, prior to a reorganization of plate boundaries and plate motion vectors, after which the Grenada basin formed. Moreover, the Grenada basin does not, as in the model, continue from STEP to STEP, but is only located in the southern 2/3 of the plate. The northern 1/3, from the latitude of Guadeloupe to the north actually underwent shortening from arc to trench during opening of the Grenada basin (Philippon et al, PLOS One 2020). So the Grenada basin was bounded only on one side by a STEP, comparable to the Lau basin that the authors argue cannot be explained by their model.

We still argue that the first order features of our model and the Caribbean cases are in good agreement. We do not dispute the details outlined above, but STEP faults are still important for the dynamics of the system and decide whether the plate will break in a new point. The rotation in the Caribbean case (and elsewhere) is interesting, and our response is that as we do not include rotation in our setup, it represents an obvious simplification of our models, but not a fatal one. The key thing is that our models are useful, provided rotation is not such a dominant effect in a natural systems that it is the first order control on spreading and ridge jumps. This does not appear to be the case in the Caribbean. We now further emphasize the tectonic complexity of the Caribbean in lines 102-105.

Likewise, there is an angle of 20° between the extension directions of the West Scotia Sea and East Scotia Sea. In my review I explained how this resulted from a reorganization of the trench due to ridge arrival, and the reconstruction on which I based those remarks was recently published in van de Lagemaat et al (Earth-Sci Rev 2021). Moreover, there is the issue that the West and East Scotia ridges were simultaneously active for at least 11 Ma: the authors argue against this now in a table in their Supplementary Data section. They write: "We here point out that the interpretation of spreading times in the East Scotia Ridge with observed chrons (e.g. Figure 4 in Eagles & Jokat (2014)) assume the East Scotia Ridge to have been formed close of the trench. The chrons can be interpreted differently, if we assume the East Scotia Ridge was formed within the basin created by the West Scotia Ridge: This leads to the chrons 3A (6-8 Ma) to be the only chron to be associated to the East Scotia ridge. The chrons interpreted as 5 and 5C in Eagles & Jokat (2014) would have been created by the West Scotia Ridge, not the East Scotia Ridge." This interpretation is undefendable: (i) the West and East Scotia Basins are separated by the Central Scotia Basin that has E-W trending magnetic anomalies, and even though the origin of these anomalies and this crust is debated (either early, trench-parallel upper plate extension, or trapped pre-subduction initiation South American lithosphere) there is a long-standing consensus that the crust of the Central Scotia Basin is older than both the West and East Scotia Basins. (ii) the anomalies in the West Scotia basin have a different orientation than in the East Scotia Basin; (iii) all anomalies older than chrons 3A are accounted for in the West Scotia Basin on either side of the ridge. So the anomalies in the East Scotia Basin were not created in the West Scotia Basin, but to the east of the Central Scotia Basin. And there is no marked change in spreading direction or rate between the oldest anomaly (C5Cn.3n.o) and the present-day, see Larter et al 2003; Fig 4 in van de Lagemaat et al 2021), so the argument that the East Scotia basin became the 'dominant' spreading center is not defensible: first there were two spreading centers, and then there was one.

We thank the reviewer for pointing out the mistake on the interpretation of the isochrones in the West and East Scotia Sea. Following the reviewer's suggestion and the published models (i.e., Eagles and Jokat, 2014), we now consider that the East Scotia ridge starts spreading at 17 Ma (and not at 6 Ma as we previously thought). We agree that there was an overlap in the ages of the spreading centres of the West and East Scotia ridges from 17 to 6 Ma – accepting the reviewer's argument, above. Eagles and Jokat (2014) also suggest that at 17 Ma spreading of the West Scotia ridge significantly decrease. We acknowledge that

this natural example in which two spreading centres are active at the same time is different from our models where there is a switch-off of spreading in the older centre when the new centre starts. But, the important thing here is that at 17 Ma a new ridge does develop in the East Scotia Sea, or, in other words, the overriding plate does break closer to trench, as it is the case in our models. So, we think that our models are still relevant, but we now highlight the greater complexity in this particular natural example (see text in the supplementary material table).

Considering the formation of the new spreading centre at 17 Ma (Figure 1, Supplementary Data revised), we recomputed the distance between the trench and the spreading centre, the duration of the spreading, and the trench length. We changed these values in the table in the supplementary material and in the plots in Fig. 4. Note that even with the new values, the data points still fit our conceptual model and the conclusions still stand.

Finally, in the Calabrian case, the hyperextended, magma-poor Marsili ridges that is more trenchward than the somewhat older Vavilov ridge are interpreted here as a 'ridge jump'. But is this a ridge jump? These hyperextended portions formed in an upper plate that consists of nappes and continental crust, and this crust has been extended since the Miocene along major detachments, from the Sardinia margin all the way to the Croton basin. At some stage, there was no continent left to stretch, and instead mantle rocks were exhumed, but the first arrival of those mantle rocks at the surface and the associated melting, which is a bit younger in Marsili than in Vavilov, do not mark the onset of upper plate extension, nor an age progression of extension towards the trench. The extension in the Croton basin in Calabria, or elsewhere in the Calabrian nappe stack, long predates the formation of Vavilov and Marsili. The model of a ridge jump in an oceanic upper plate does not compare to the Tyrrhenian basin.

In their response, the authors write: "In detail, every natural system will be more complicated than our model set-up. But, like the Scotia and Tyrrhenian Sea cases, it is striking how the internal complexity of each region does not mask the essential point that back-arc ridge jumps do occur in many systems, and that these are explained by our model without the need to reach for an ad hoc explanation in each case." Perhaps there is no need to 'reach for an ad hoc explanation', but in all cases discussed by the authors, the kinematic history simply shows that the relocation of the accommodation of extension coincided with reorganizations of the plate boundary (Caribbean, Scotia), or that the 'jump' is not a jump, but simply a different moment of exhumation of mantle rock after a prolonged, distributed extension history. These reconstructions were not made to find an ad hoc explanation for ridge jumps, the explanations simply follow from the sequence of events.

The ages of magmatism clearly show that magmatism in Marsili is younger than in Vavilov (2-0 Ma vs. 5-2 Ma) (e.g. Faccenna et al., 2004). Therefore, there is a jump in the location of magmatism towards the trench. Also, the Croton basin is not in the back-arc, so the timing of its extension is irrelevant in this context. Nor are we convinced that the Marsili Basin has extended since the Miocene. Finally for this region, we reject the characterisation of the tectonics as a "prolonged, distributed extension history": the progression of rifting is much more discrete than this phrase suggests.

So what does this modeling study really solve? What is the importance of the conclusion that in some very specific cases, ridges may jump because transforms become too long?

The authors formulate this importance as: "Our models explain why narrow subducting plates (e.g. Calabrian slab), have more frequent and closely-spaced spreading jumps than wider subduction zones (e.g. Scotia). It also explains why wide back-arc basins undergo no spreading centre jumps in their life cycle." But the back-arc basins that are discussed are few, and have one, maybe two ridge jumps, and 'frequency', 'closely-spaced', 'narrow' and

'wider' are not quantified in this paper, and I don't think they can be quantified or substantiated. Second, even if the conclusion is valid, what is so important about this?

We argue that our models capture first order features of back-arc ridge jumps, and that each of the three natural examples described above fits the main features of the models. We have explained where there are differences between the models and natural examples (e.g. plate rotations), but these do not invalidate the concepts. It's worth highlighting how other basins are included as natural examples – see Figure 4, for the cases where these wider basins have not generated ridge jumps, consistent with our model. The models are quantified; we have set-up examples with specific parameters, so that when we use a term like “narrow” it can be related for the 400 km set-up, for example. We reach a simple but quantitative conclusion, summarised in our abstract “Spreading centres jump towards their subduction zone if the distance from trench to spreading centre becomes too long, ~1.3 times plate width”. Our review of natural examples is also quantitative – see the division in behaviour shown in Figure 4. The importance of this work is multifold: for one, we quantify a poorly constrained process in nature, thereby explaining its physics. There is also clear wider relevance of this work: e.g., the fit of models to natural examples (Figure 4) can be used to provide new and much needed constraints on the rheology of lithosphere, which is particularly welcome given the uncertain influence of hydration from the nearby subduction zone. The supplementary material section “The relationship between transform fault lengths, overriding plate widths, and the timing of back-arc spreading centre jumps” elaborates the energy balance behind the ridge jumps, which directly relates to the strength of the lithosphere and the frictional resistance of the transform faults. This is now further explained in the paragraph on the energy balance for the ridge jump.

I am sorry to be so critical, and I am well aware that reality is always more complex than model setups. But perhaps the authors can explain what new opportunities are created by understanding why ridges between transforms above subduction zones jump, and how that will help us to advance predictability of geodynamic systems?

The process of back-arc ridge jump is enigmatic and classical plate tectonics cannot explain it. That means that we do not fully understand the subduction systems that drives plate tectonics. As already explained with the previous comment, our work offers a way forward in understanding this process, showing that subduction and associated back-arcs are strongly transient, and quantifying the rheology of the upper plate and the STEP fault strength, and this has been added in the section “The relationship between transform fault lengths, overriding plate widths, and the timing of back-arc spreading centre jumps. “.

Utrecht, May 31, 2021
Douwe van Hinsbergen

Reviewer #2 (Remarks to the Author):

The Authors properly revised the paper that I can now recommend for publication.
Taras Gerya, 04.06.2021, Zurich

We thank the reviewer for their previous comments that significantly strengthened this manuscript.

Reviewer #3 (Remarks to the Author):

Review of revised manuscript "Episodic back-arc spreading centre jumps controlled by transform fault to overriding plate strength ratio" submitted to Nature Communications by Nicholas Schliffke and co-authors.

In my opinion, the authors have improved the manuscript sufficiently according to the reviewer comments to warrant its publication in the journal.

Kind regards,

Vincent Strak

We thank the reviewer for their previous comments that significantly strengthened this manuscript.